# Extraocular, rod-like photoreceptors in a flatworm express xenopsin photopigment

Kate A Rawlinson[1,2,3]*, Francois Lapraz[4], Edward R Ballister[5,6], Mark Terasaki[3,7], Jessica Rodgers[6], Richard J McDowell[6], Johannes Girstmair[8,9], Katharine E Criswell[2,3], Miklos Boldogkoi[6], Fraser Simpson[8], David Goulding[1], Claire Cormie[1], Brian Hall[10], Robert J Lucas[6], Maximilian J Telford[8]

[1]Wellcome Sanger Institute, Hinxton, United Kingdom; [2]Department of Zoology, University of Cambridge, Cambridge, United Kingdom; [3]Marine Biological Laboratory, Woods Hole, United States; [4]Université Côte D'Azur, CNRS, Institut de Biologie Valrose, Nice, France; [5]New York University School of Medicine, New York, United States; [6]Division of Neuroscience and Experimental Psychology, Faculty of Biology, Medicine and Health, University of Manchester, Manchester, United Kingdom; [7]University of Connecticut Health Center, Farmington, United States; [8]Centre for Life's Origins and Evolution, Department of Genetics, Evolution and Environment, University College London, London, United Kingdom; [9]Max Planck Institute of Molecular Cell Biology and Genetics, Dresden, Germany; [10]Department of Biology, Dalhousie University, Halifax, Canada

*For correspondence:
kr16@sanger.ac.uk

Competing interests: The authors declare that no competing interests exist.

**Abstract** Animals detect light using opsin photopigments. Xenopsin, a recently classified subtype of opsin, challenges our views on opsin and photoreceptor evolution. Originally thought to belong to the $G\alpha i$-coupled ciliary opsins, xenopsins are now understood to have diverged from ciliary opsins in pre-bilaterian times, but little is known about the cells that deploy these proteins, or if they form a photopigment and drive phototransduction. We characterized xenopsin in a flatworm, *Maritigrella crozieri*, and found it expressed in ciliary cells of eyes in the larva, and in extraocular cells around the brain in the adult. These extraocular cells house hundreds of cilia in an intra-cellular vacuole (phaosome). Functional assays in human cells show *Maritigrella* xenopsin drives phototransduction primarily by coupling to $G\alpha i$. These findings highlight similarities between xenopsin and c-opsin and reveal a novel type of opsin-expressing cell that, like jawed vertebrate rods, encloses the ciliary membrane within their own plasma membrane.
DOI: https://doi.org/10.7554/eLife.45465.001

## Introduction

Light is a key biological stimulus for most animals, and a rich diversity of photosensitive cells has evolved. Depending on the form of their elaborated apical plasma membranes, these cells have been characterized as either ciliary photoreceptors (CPRs) or rhabdomeric (microvillar) photoreceptors (RPRs) (*Eakin, 1982*). When rhabdomeric and ciliary photoreceptors coexist in the same organism, one type (rhabdomeric in most invertebrates, ciliary in vertebrates) typically dominates in the eyes while the other performs nonvisual functions in the eyes or is present as extraocular photoreceptors (*Arendt et al., 2004*; *Yau and Hardie, 2009*). Photopigments are responsible for the light-dependent chemical reactions in these cells, and all animal phyla, with the exception of sponges, employ photopigments composed of opsin-class G-protein-coupled receptors (GPCRs) coupled with a light-sensitive chromophore (a retinaldehyde) (*Nilsson, 2013*; *Bok et al., 2017*). The initial characterization of opsins in bilaterian animals identified several conserved opsin gene families

**eLife digest** Eyes are elaborate organs that many animals use to detect light and see, but light can also be sensed in other, simpler ways and for purposes other than seeing. All animals that perceive light rely on cells called photoreceptors, which come in two main types: ciliary or rhabdomeric. Sometimes, an organism has both types of photoreceptors, but one is typically more important than the other. For example, most vertebrates see using ciliary photoreceptors, while rhabdomeric photoreceptors underpin vision in invertebrates.

Flatworms are invertebrates that have long been studied due to their ability to regenerate following injuries. These worms have rhabdomeric photoreceptors in their eyes, but they also have unusual cells outside their eyes that have cilia – slender protuberances from the cell body - and could potentially be light sensitive. One obvious way to test if a cell is a photoreceptor is to see if it produces any light-sensing proteins, such as opsins. Until recently it was thought that each type of photoreceptor produced a different opsin, which were therefore classified into rhabdomeric of ciliary opsins. However, recent work has identified a new type of opsin, called xenopsin, in the ciliary photoreceptors of the larvae of some marine invertebrates.

To determine whether the cells outside the flatworm's eye were ciliary photoreceptors, Rawlinson et al. examined the genetic code of 30 flatworm species looking for ciliary opsin and xenopsin genes. This search revealed that all the flatworm species studied contained the genetic sequence for xenopsin, but not for the ciliary opsin.

Rawlinson et al. chose the tiger flatworm to perform further experiments. First, they showed that, in this species, xenopsin genes are active both in the eyes of larvae and in the unusual ciliary cells found outside the eyes of the adult. Next, they put the xenopsin from the tiger flatworm into human embryonic kidney cells, and found that when the protein is present these cells can respond to light. This demonstrates that the newly discovered xenopsin is light-sensitive, suggesting that the unusual ciliary cells found expressing this protein outside the eyes in flatworms are likely photoreceptive cells.

It is unclear why flatworms have developed these unusual ciliary photoreceptor cells or what their purpose is outside the eye. Often, photoreceptor cells outside the eyes are used to align the 'body clock' with the day-night cycle. This can be a factor in healing, hinting perhaps that these newly found cells may have a role in flatworms' ability to regenerate.

DOI: https://doi.org/10.7554/eLife.45465.002

(*Terakita, 2005*), and each family has been associated with distinct photoreceptor cell types and specific downstream G-protein phototransduction cascades (reviewed in *Lamb, 2013*). For example, ciliary (c)-opsins are expressed in ciliary photoreceptor cells where they typically activate members of the Gαi family (including Gαi, Gαo and Gαt), while rhabdomeric (r)-opsins are expressed in rhabdomeric photoreceptors and activate Gαq family members (*Shichida and Matsuyama, 2009*). The recent accumulation of sequence data from a taxonomically broad set of animals has, however, revealed a far greater diversity of opsins (*Porter et al., 2012*; *Ramirez et al., 2016*; *Bok et al., 2017*; *Vöcking et al., 2017*), and the ability to localize the mRNA transcripts and proteins of opsins in a diversity of animals has uncovered many new and morphologically divergent photosensitive cell types, both ocular and extraocular (*Vöcking et al., 2017*; *Bok et al., 2017*).

The recent identification of one novel group of opsins – the xenopsins (*Ramirez et al., 2016*) – is leading to a reevaluation of eye and photoreceptor cell type evolution (*Vöcking et al., 2017*). Xenopsins have been found in several lophotrochozoan phyla: molluscs, rotifers, brachiopods, flatworms and an annelid (*Ramirez et al., 2016*; *Vöcking et al., 2017*). They share with some ciliary opsins a characteristic c-terminal sequence motif (NVQ) and were originally classified as part of the c-opsins (*Passamaneck et al., 2011*; *Albertin et al., 2015*; *Yoshida et al., 2015*). All recent opsin phylogenies have, however, shown xenopsins to be phylogenetically distinct from c-opsins (*Ramirez et al., 2016*; *Bok et al., 2017*; *Vöcking et al., 2017*; *Quiroga Artigas et al., 2018*) and gene structure analysis supports this distinction (*Vöcking et al., 2017*). The relationship between xenopsins (lophotrochozoan protostome specific) and c-opsins (which are found in protostomes and deuterostomes) suggests that both opsins were present in the last common ancestor of Bilateria, and that xenopsins

were subsequently lost in deuterostomes and ecdysozoan protostomes (*Vöcking et al., 2017*). Existing data on the expression of xenopsins are limited to the larval stages of a chiton and a brachiopod. In the brachiopod, *xenopsin* is expressed in cells with elaborated cilia and shading pigment that is pigmented eyespots (*Passamaneck et al., 2011*), whereas, unusually, in the chiton larva it is co-expressed with *r-opsin* in cells containing both cilia and microvilli. Some of these cells are supported by pigmented cells so they probably form simple eyes, whereas others lack pigment and cannot act as visual photoreceptors (*Vöcking et al., 2017*).

While the presence of xenopsins in cells with elaborated ciliary surfaces and their association with pigmented cells is strongly suggestive of a role for xenopsins in photoreception, this function has not yet been demonstrated. Furthermore, if xenopsins do detect light, the subsequent phototransduction pathway is unknown. Determining these factors, and better understanding the phylogenetic distribution of xenopsins and of the cells in which they are expressed is essential for understanding the evolution of this opsin subtype and of the photoreceptors that use them (*Arendt, 2017*).

Flatworms (Platyhelminthes) are one of the most diverse and biomedically important groups of invertebrates (*Laumer et al., 2015*). Their eyes typically consist of photoreceptors with rhabdomes of microvilli that are associated with pigmented shading cells (*Sopott-Ehlers et al., 2001*) and which, in planarian triclad flatworms, have been shown to express rhabdomeric opsin (*Sánchez Alvarado and Newmark, 1999*; *Pineda et al., 2000*) and conserved members of the r-opsin signaling cascade (e.g. Gαq, Trp channel-encoding genes) (*Lapan and Reddien, 2012*). The presence and nature of ciliary photoreceptors in flatworms is still unclear but the description of xenopsins (but not c-opsins) in flatworms (*Vöcking et al., 2017*) suggests CPRs may exist. Furthermore, ultrastructural studies have identified cells with elaborated ciliary membranes - putative CPRs (*Sopott-Ehlers, 1991*; *Lyons, 1972*; *Kearn, 1993*; *Rohde and Watson, 1991*) - but these have not been studied at the molecular level. In larvae of the polyclad *Pseudoceros canadensis*, ultrastructural studies identified three different types of CPR; the epidermal eyespot – a pigmented epidermal cell with elaborated ciliary membranes (*Lanfranchi et al., 1981*; *Eakin and Brandenburger, 1981*); a cerebral eye consisting of a CPR adjacent to RPRs cupped by a supporting pigmented cell (*Eakin and Brandenburger, 1981*); and distinct extraocular cells in the epidermis possessing multiple cilia projecting into an intra-cellular vacuole (*Lacalli, 1983*) known as a phaosome (*Fournier, 1984*). This phaosomal cell type has been found in all classes of flatworm (except triclads and bothrioplanids) (*Sopott-Ehlers et al., 2001*; *Fournier, 1984*), and the distinct morphology led to the suggestion that they are a derived feature of flatworms (*Sopott-Ehlers et al., 2001*).

Here we analyze the localization of a xenopsin protein in a polyclad flatworm at two developmental stages; the newly hatched larva and, for the first time, in an adult lophotrochozoan. In the adult we find xenopsin[+], extraocular, ciliary cells and we characterize their distinct ultrastructure, revealing that, like jawed vertebrate rod photoreceptors, they enclose their cilia inside their own plasma membrane. We then carry out the first functional cellular assays on a xenopsin, exploring whether polyclad xenopsin can form a photopigment, and which classes of Gα protein it can couple to (the first step in phototransduction cascades). Together our findings show similarities between xenopsin and c-opsin and provide the first molecular and functional evidence of ciliary photoreceptors in flatworms.

## Results

### Xenopsins and rhabdomeric opsins in flatworms

A 346 amino acid gene product showing similarity to protostome c-opsin and to xenopsin was predicted from a *Maritigrella crozieri* transcriptome contig using BLAST (*Madden, 2002*). Opsins showing similar degrees of similarity were found in transcriptomes from five other flatworm taxa (three polyclads; *Prostheceraeus vittatus, Stylochus ellipticus, Leptoplana tremellaris* and two triclad species *Schmidtea mediterranea, Dendrocoelum lacteum*). We did not find homologous sequences in the remaining 24 flatworm species representing other flatworm classes, including those in which putative CPRs have been described (catenulids, macrostomids, rhabdocoels, proseriates, monogeneans, cestodes and trematodes). Searching more broadly we found additional opsins similar to protostome c-opsins and xenopsins in a bryozoan, *Bugula nerita*, and in a chaetognath, *Pterosagitta draco*.

In our phylogenetic analyzes of these putative flatworm, bryozoan and chaetognath opsins in the context of the metazoan opsin gene family, all cluster with xenopsins (*Figure 1*; *Figure 1—figure supplement 1*). Several polyclad flatworm species show xenopsin paralogs distributed across two xenopsin subgroups (*Vöcking et al., 2017*); our six polyclad and triclad sequences all group with clade A and we have found xenopsins from three additional taxa to be included in this clade; *Maritigrella crozieri*, *Dendrocoelum lacteum*, *Leptoplana tremellaris* (*Figure 1*; *Figure 1—figure supplement 1*). The xenopsins are a well-supported monophyletic group of genes most closely related to cnidopsins. The xenopsin/cnidopsin group is sister to the tetraopsins and all are part of a larger clade including bathyopsins and canonical c-opsins (*Figure 1*).

Support for the relationships between these well-defined opsin subtypes is very low, indicating that these relationships should be interpreted cautiously. The need for caution is reinforced by the observation that removing the smaller opsin clades from our dataset (chaopsins, bathyopsins, ctenophore and anthozoan opsins), changes the topology of the deeper nodes of our trees (*Figure 1—figure supplement 2*).

The flatworm xenopsin protein sequences possess seven transmembrane domains (characteristic of all G protein-coupled receptors) as well as a conserved lysine in transmembrane domain VII, which is specific to opsins and which forms a Schiff base with the retinal chromophore to form a photopigment (*Figure 1—figure supplement 3*). *Mc* xenopsin also possesses a tripeptide motif, NxQ, at positions 310–312, which is reported in other xenopsins (*Passamaneck et al., 2011*; *Vöcking et al., 2017*) and in ciliary opsins where it is known to be crucial for G-protein activation (*Marin et al., 2000*; *Gühmann et al., 2015*).

We have found that a second amino acid signature, VxPx, found in vertebrate ciliary opsins at positions 423–426 is also present in xenopsins as well as in ciliary opsin sequences from non-vertebrate chordates (tunicate and lamprey), annelid c-opsins and cnidarian cnidopsins (*Figure 1—figure supplement 3*). In c-opsins this motif directly binds the small GTPase Arf4 to direct vertebrate rhodopsin (a ciliary opsin) to the primary cilia (*Deretic et al., 2005*). The presence of this motif in some ciliary opsins, xenopsins and cnidopsins suggests that Arf4 may be a shared mechanism for the active delivery of these opsins to the cilia in CPRs.

A 422 amino acid gene product related to rhabdomeric opsin was also predicted from a *Maritigrella crozieri* transcriptome contig. Nine more flatworm rhabdomeric-like opsins were predicted from both free-living and parasitic species. They possess a tripeptide motif (HP[K|R]) (supplementary figure 3) following the transmembrane helix VII, which is critical for G-protein binding in r-opsins (*Plachetzki and Oakley, 2007*). In our phylogenetic analysis, *Maritigrella r-opsin* and all flatworm *r-opsin* sequences (except one from the liver fluke *Clonorchis sinensis*) fall in a monophyletic group containing lophotrochozoan and ecdysozoan r-opsins, and deuterostome melanopsins (*Figure 1*; *Figure 1—figure supplement 1*).

Our reconstructions of the opsin gene family have resolved *Maritigrella* genes as orthologs of xenopsins and r-opsins (*Figure 1*), and we designated these genes as *Maritigrella crozieri xenopsin* (*Mc-xenopsin*) and *Maritigrella crozieri rhabdomeric opsin (Mc-r-opsin)*.

## In the larval stage xenopsin and r-opsin are expressed in eyes

In the larval stage, xenopsin protein is expressed in the epidermal eye and one of the two cerebral eyes, but not in the epidermal ciliary phaosomes (*Figure 2A–C*). *R-opsin* mRNA is expressed in both cerebral eyes (*Figure 2D*), but not in the epidermal eye (*Figure 2—figure supplement 1F*). TEM images show that the epidermal eye and one of the cerebral eyes house cilia (*Figure 2E & F*), whereas the other cerebral eye contains just microvilli (*Figure 2G*). In the epidermal eye the cilia are stacked into lamellae, whereas in the cerebral eye the cilia are unmodified and project into the opticoel. Xenopsin is co-localised with acetylated tubulin on the cilia of the cerebral eye (*Figure 2C*), showing, for the first time, xenopsin protein localization to cilia. TEM images confirm that the acTub[+] cells in the epidermis of *Maritigrella* larvae (*Figure 2B*) are cells with multiple cilia projecting into an intracellular vacuole (or phaosome) (*Figure 2H*); although thought to be CPRs (*Lacalli, 1983*; *Rawlinson, 2010*), these cells did not express xenopsin.

The epidermal eye develops before the cerebral eyes in *M.crozieri* (*Rawlinson, 2010*) and xenopsin is expressed in this eyespot during embryogenesis (*Figure 2—figure supplement 1*). The ontogenetic fates of the epidermal eye and the xenopsin expressing ciliary cell in the cerebral eye are

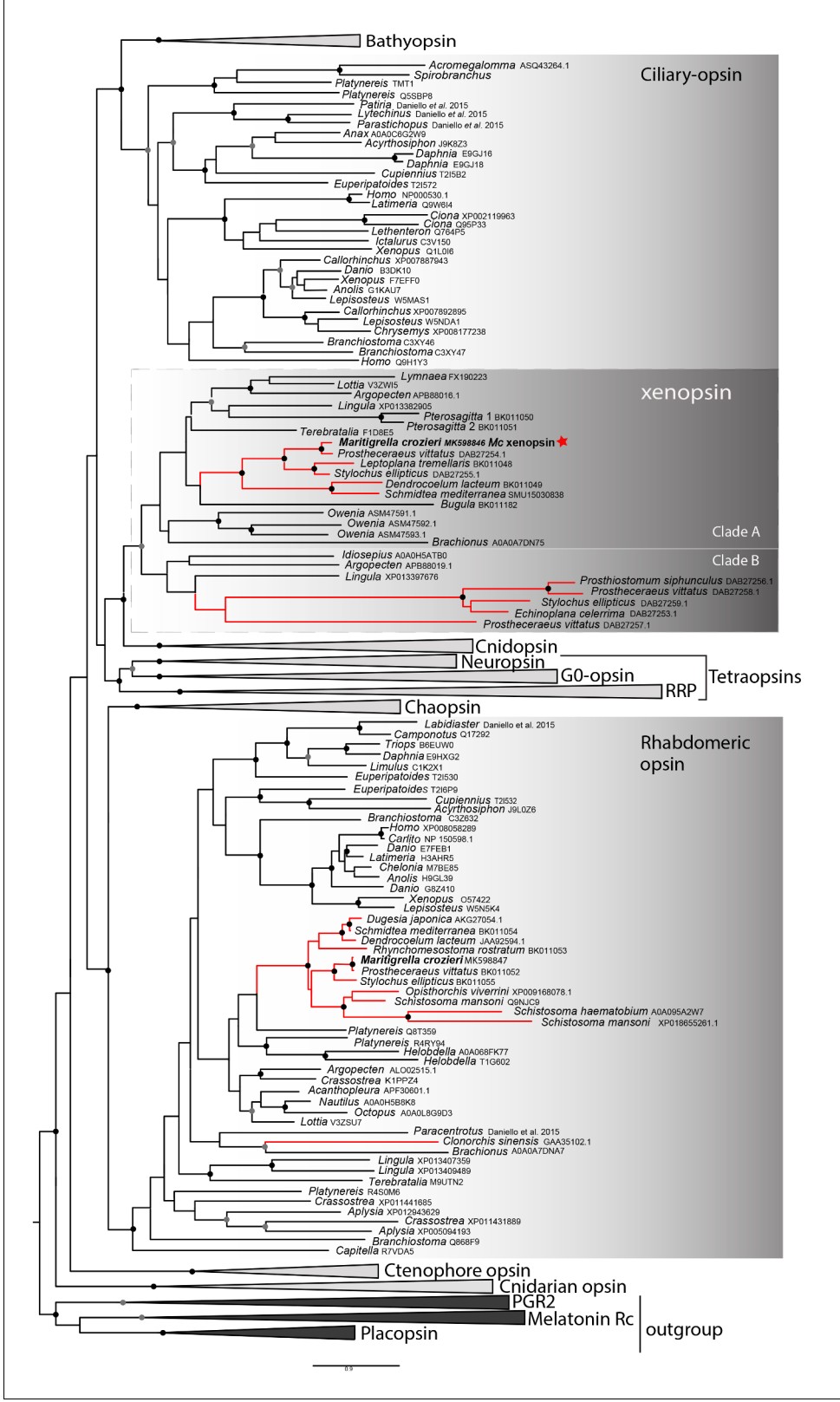

**Figure 1.** Phylogenetic analysis of metazoan opsins supports flatworm ciliary-like opsins as xenopsins and confirms a clade of flatworm rhabdomeric opsins. Support for nodes is calculated using 1000 Ultrafast bootstrap replications as well as 1000 SH-aLRT replicates and approximate aBayes single Branch testing. Black dots indicate nodes with support values for three tests ≥ 95% (0.95 for SH-aLRT replicates). Gray dots indicate nodes with support values for three tests ≥ 90% (0.90 for SH-aLRT replicates). Scale bar unit for branch length is the number of substitutions per site. Branches in red

*Figure 1 continued on next page*

*Figure 1 continued*

correspond to flatworm opsin sequences. See *Figure 1—figure supplement 1* for uncollapsed tree and *Figure 1—source data 1* for gene accession numbers. The new xenopsin sequences we found in polyclad and triclad flatworms, plus a bryozoan and chaetognath, all fall within clade A of the xenopsins.

DOI: https://doi.org/10.7554/eLife.45465.003

The following source data and figure supplements are available for figure 1:

**Source data 1.** Sequence data for opsins used in phylogenetic analyses for *Figure 1*.

DOI: https://doi.org/10.7554/eLife.45465.007

**Figure supplement 1.** Uncollapsed tree of IQ-TREE phylogenetic reconstruction of opsin relationships.

DOI: https://doi.org/10.7554/eLife.45465.004

**Figure supplement 2.** IQtree and RaxML trees showing the influence of the small opsin clades (i.e. chaopsins, bathyopsins, ctenophore and anthozoan opsins) on the position of xenopsins in relation to c-opsins and tetra-opsins (Neuropsin, Go-opsin and RRP); inclusion of these small opsin clades brings xenopsins close to tetraopsins (full dataset), their exclusion brings xenopsins close to c-opsins (reduced dataset).

DOI: https://doi.org/10.7554/eLife.45465.005

**Figure supplement 3.** Alignment of major opsin clades showing conserved lysine in transmembrane domain VII, which binds to the retinal chromophore to form a photopigment.

DOI: https://doi.org/10.7554/eLife.45465.006

unknown; it could be that both are transient larval features, as all pigmented eyes in adult *Mariti-grella* are sub-epidermal and express *r-opsin,* but not xenopsin (Figure 5F).

## In the adult, xenopsin is expressed in extraocular ciliary phaosomes and r-opsin is expressed in the eyes

No putative ciliary photoreceptors have been documented in an adult polyclad so, to identify candidates in *Maritigrella* we first used antibodies against acetylated tubulin and discovered two clusters of up to 100 acTub+ cells, one either side of the brain (*Figure 3B,Bi and Di*). The cells are distributed from the anterior to the posterior of the brain (*Figure 3Di*) and extend laterally above nearby branches of the intestine (*Figure 3B* and Figure 5B). Histological staining showed that these cells are embedded in extracellular matrix outside of, and lateral to, the brain capsule (*Figure 3C and Ci*) and that they sit in close proximity to the main nerve tracts (*Figure 3Ci*). The cells appear stalked with a nucleus at one end and a balloon-shaped phaosome in the outer segment at the opposite end, which houses multiple cilia (*Figure 3E,Ei*).

TEM and serial SEM analyzes of these acTub+ cells showed that they house multiple, unmodified cilia in an intra-cellular vacuole and that they sit in close proximity to each other, in dense aggregations (*Figure 4A and B*, and *Figure 4—video 1*). These cells are not associated with any pigmented supporting cells, that is they are extraocular and, although there are unpigmented cells in close proximity (*Figure 4*), it seems as though the intra-cellular vacuole is completely enclosed by the cell itself and can therefore be considered a phaosome (*Figure 4—video 2*).

The intracellular cavities/phaosomes have diameters up to 23 µm and the wall of the cavity is comparatively thin in certain areas (~40 nm). The cytoplasm bordering the internal cavity contains mitochondria (*Figure 4C and D*), and the cilia are anchored in the cytoplasmic layer by basal bodies (*Figure 4D*), each basal body gives rise to one cilium. Counting the basal bodies from the serial SEM of a single phaosome reveals at least 421 cilia projecting into the phaosome (*Figure 4—video 2*). The cilia are unbranched and emerge all around the diameter of the cavity (*Figure 4E*, *Figure 4—videos 3* and *4*), forming a tightly intertwined bundle (*Figure 4B*). They have an average diameter of 0.2 µm and length of 7.2 µm. This represents a total membrane surface area per phaosome of approximately 600 µm².

The cilia are generally orientated horizontally in relation to the dorso-ventral body axis and, in some of the phaosomes, the cilia appear to be arranged in a spirally coiled bundle (*Figure 4—video 1*). Near their bases, the axonemata show a $9 \times 2 + 2$ arrangement of microtubules. With increasing distance from the base, $9 + 2$ singlets are encountered and then the nine-fold symmetry becomes disorgani zed and microtubular singlets are found (*Figure 4F*; *Figure 4—video 3*). As no ciliary root-lets were evident it is most likely that these cilia are non-motile; however, possible dynein arms (gen-erally associated with motile cilia) were observed attached to the A-tubules near the bases of the

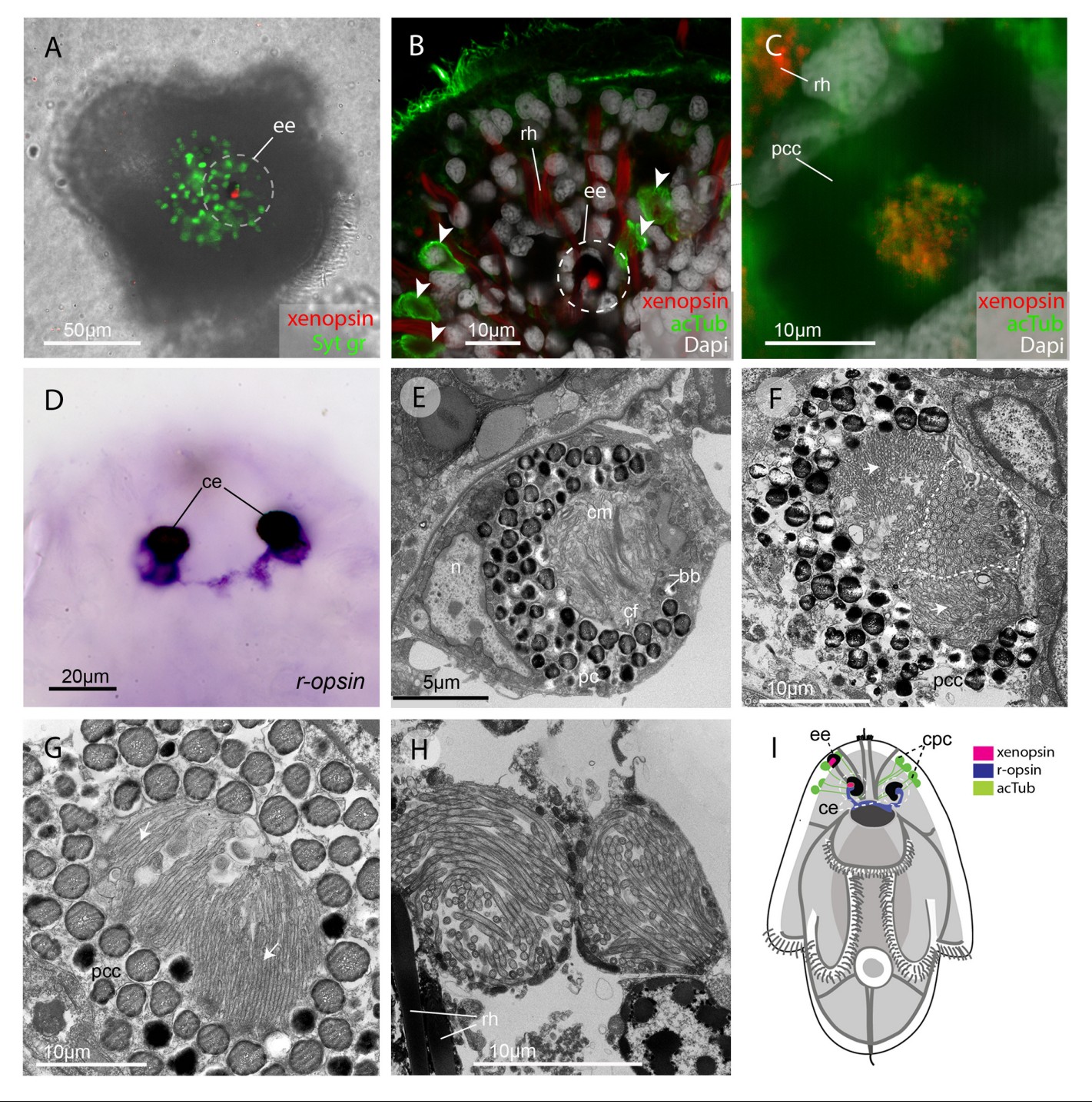

**Figure 2.** In the larva (1 day post-hatching) xenopsin protein is localized to ciliary cells in the eyes. (A) Apical view of larva showing xenopsin (red) in the epidermal eye (*ee*) (100% of individuals examined, n = 50) (OpenSPIM image, Syt gr = Sytox green, staining nuclei and bright-field image also reveal photoresence pigments). (B) Confocal optical section showing xenopsin in the epidermal eye (circled) but not in the acetylated tubulin⁺ (acTub) cells in the epidermis (arrowheads); autofluoresence of rhabdites (*rh*). (C) Xenopsin is co-localised with acTub in one of the two cerebral eyes providing evidence that xenopsin protein localizes to cilia (this varies between the right and left cerebral eye in different larvae, 50:50, n = 10)(pcc = pigment cup cell). (D) *R-opsin* is expressed in both cerebral eyes (*ce*). (100% of individuals examined, n = 30), (E) TEM image showing the epidermal eye which houses elaborated ciliary membranes (*cm*) inside a pigment cup (*pc*)(basal bodies, *bb*; cross section of ciliary flagella, *cf*, nucleus,*n*) (100% of individuals examined, n = 3). (F) Ultrastructure of a cerebral eye showing cilia (inside dashed line) and microvilli (arrows) cupped within a pigment cell (*pcc*). (G) Ultrastructure of another cerebral eye showing microvilli (arrows) cupped inside a pigment cup cell. (H) Multiple cilia projecting into phaosomes (intra-

*Figure 2 continued on next page*

*Figure 2 continued*

cellular vacuoles) in the epidermis. (I) A schematic of a larva summarizing the expression of xenopsin, *r-opsin* and acTub. In the larva xenopsin is expressed in two of the three putative ciliary photoreceptor cell types: the epidermal eye, a cerebral eye, but not in the ciliary phaosome cells (*cpc*).
DOI: https://doi.org/10.7554/eLife.45465.008

The following figure supplement is available for figure 2:

**Figure supplement 1.** Opsin localization and expression in Maritigrella embryos and larva.
DOI: https://doi.org/10.7554/eLife.45465.009

cilia (*Figure 4F*). We observed these cells in live adults and although the cilia inside the phaosomes were visible, no cilia were seen moving, even in response to changes in illumination.

These cells are similar in ultrastructure to the extraocular ciliary phaosome cells in the epidermis of the larva (*Figure 2H*). Unlike those in the larval stage, however, in the adult, *Mc*-xenopsin was strongly co-localised with acetylated tubulin in these cells (*Figure 5A–E*). The xenopsin protein was located throughout the cytoplasm of the cell with strong expression around the base of the cilia (*Figure 5Di and E*). These xenopsin$^+$ cells sit ventro-lateral to the cerebral eyes that themselves consist of pigmented cup cells and *r-opsin*$^+$ cells that extend down to the brain (*Figure 5F*). The

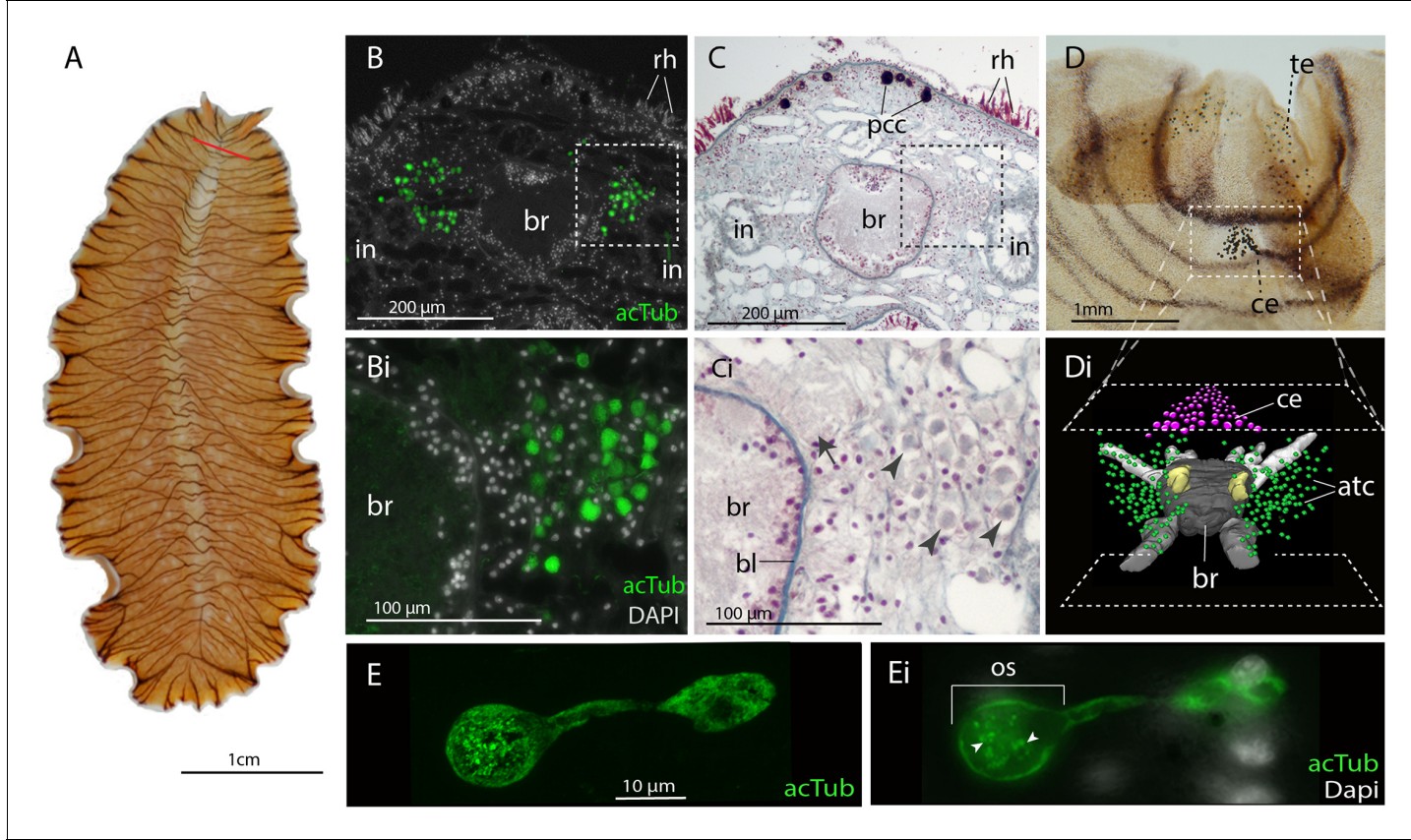

**Figure 3.** Acetylated tubulin staining identifies two dense clusters of extraocular cells, possible ciliary photoreceptors (CPR), either side of the adult brain. (A) Live adult, red line shows plane of cross section in B-C. (B and C) Consecutive sections showing; (B) two clusters of acetylated tubulin$^+$ cells and; (C) their distribution between the brain (*br*) (which is encapsulated in a basal lamina, *bl*) and intestinal branches (*in*) (n = 5 individuals). Bi and Ci) Close up showing that these cells (arrowheads) are embedded in extracellular matrix in close proximity to the main nerve tracts (arrow). Pigment cup cells (*pcc*), rhabdites (*rh*). (D) Anterior end of adult showing pigmented eyes above the brain (cerebral eyes, *ce*) and on the tentacles (tentacular eyes, *te*). (Di) Schematised distribution of acetylated tubulin$^+$ cells (*act*) and cerebral eyes on a micro-CT reconstruction of the brain and main anterior (white), posterior (gray), and two of the optic (yellow) nerve tracts. (E) Confocal projection of a putative CPR and, Ei) an optical slice of the same cell showing cilia projecting into the intra-cellular vacuole or phaosome (arrowheads) in the outer segment (os).
DOI: https://doi.org/10.7554/eLife.45465.010

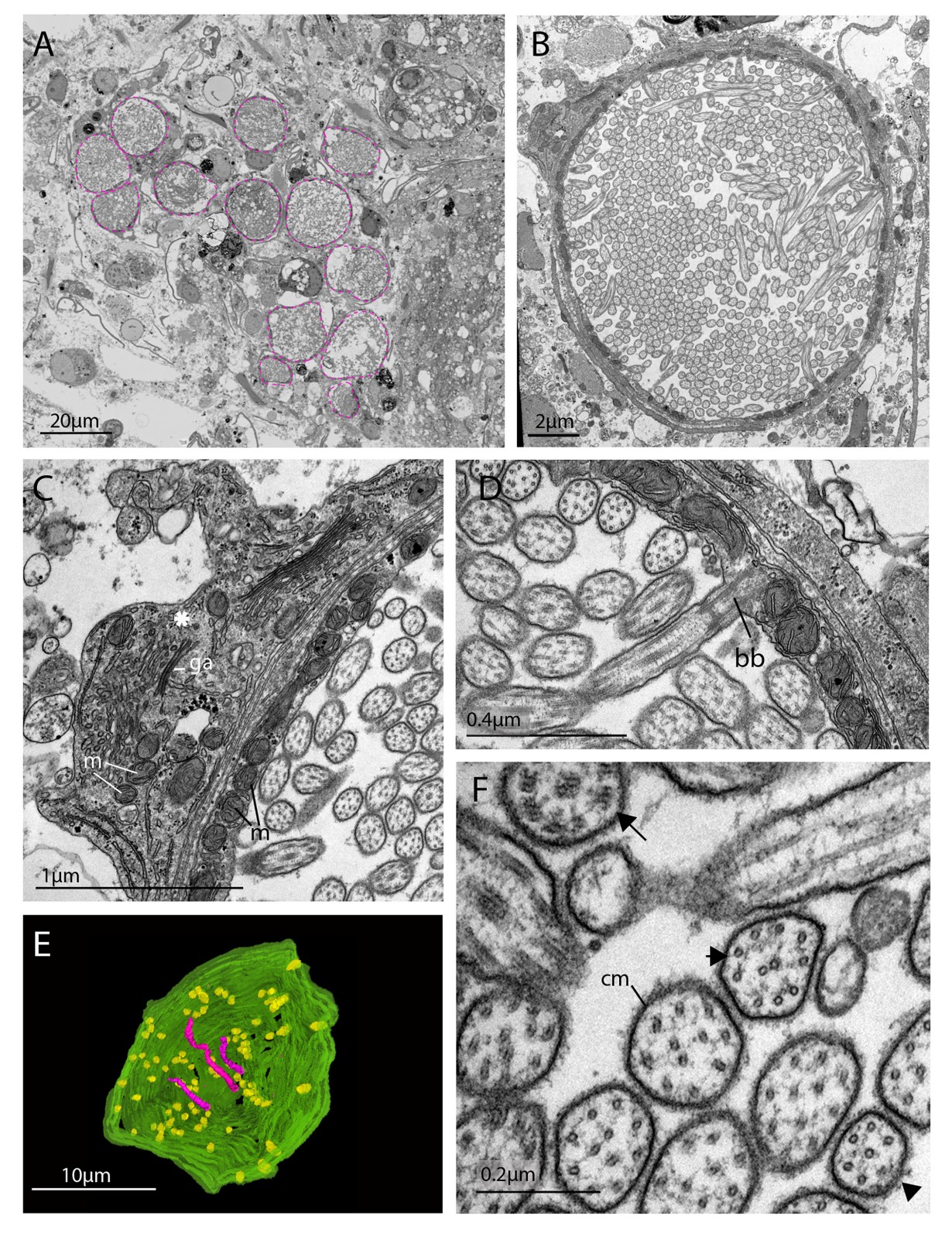

**Figure 4.** The morphology of the ciliary phaosomal cells in adult *Maritigrella crozieri*. (**A**) A dense cluster of intra-cellular vacuoles (phaosomes – highlighted in magenta) filled with cilia (n = 3 individuals). (**B**) Multiple cilia in the phaosome. (**C**) A possible unpigmented supporting cell (asterisk) wrapping around the phaosome cell with mitochondria (m) and Golgi apparatus (ga) in the cytoplasm. (**D**) Ciliary axonemata (ax) are anchored in the cytoplasmic layer (cl) by basal bodies (bb). (**E**) 3D reconstruction of the interior of a third of a phaosome, showing that the cilia are unbranched (pink)

*Figure 4 continued on next page*

*Figure 4 continued*

and the basal bodies (yellow) are distributed all around the phaosome. (F) Cross sections of the ciliary axonemata show various arrangements of microtubules: $9 \times 2^+$ two with dynein arms attached to the A-tubules (arrow), 9 + 2 singlets (double arrowheads), and singlets (triple arrowheads). This variation is related to the distance from the basal body (*Figure 4—videos 3*), (cm) ciliary membrane.

DOI: https://doi.org/10.7554/eLife.45465.011

The following videos are available for figure 4:

**Figure 4—video 1.** Serial SEM images (101 × 500 nm sections = 50.5 μm total thickness) showing a cluster of ciliary phaosomes (intra-cellular vacuoles housing multiple cilia) that form the outer segment of the putative extraocular CPR cells in adult *Maritigrella*.

DOI: https://doi.org/10.7554/eLife.45465.012

**Figure 4—video 2.** Serial SEM images (47 × 400 nm sections = 18.8 μm total thickness) showing a complete phaosome.

DOI: https://doi.org/10.7554/eLife.45465.013

**Figure 4—video 3.** Serial SEM images (72 x 100nm sections = 7.2μm total thickness) showing a third of a phaosome.

DOI: https://doi.org/10.7554/eLife.45465.014

**Figure 4—video 4.** A 3D reconstruction of series in *Figure 4—video 3* shows that the cilia (pink) are unbranched and basal bodies (yellow) emerge all around the diameter of the cavity.

DOI: https://doi.org/10.7554/eLife.45465.015

pigmented eyes on the tentacles (*Figure 3D*) also express *r-opsin* (*Figure 5F*). The non-overlapping expression of xenopsin and *r-opsin* indicates that these opsins are expressed in two distinct cell types, with *r-opsin* expressed in the eyes and xenopsin expressed in the extraocular cells (*Figure 5G*). As is typical for *r-opsin*-expressing cells, they also express Gαq, as revealed by antibody staining (*Figure 5H*). As there was no prior information on which classes of Gα xenopsin couples with, we searched for Gα subunits in the *Maritigrella crozieri* transcriptome and identified orthologs of Gαi, Gαo and three paralogs of Gαs and Gαq (*Figure 5 – Figure 5—figure supplement 1A*; *Figure 5—source data 1*). Attempts to visualize the expression of these transcripts using in situ hybridization on larval wholemounts and adult paraffin sections were unsuccessful. There is substantial sequence conservation between *Maritigrella* and human Gα subunits (*Figure 5 – Figure 5—figure supplement 1B*), so we attempted to investigate Gα expression in *Maritigrella* using commercially available antibodies against human Gαq, Gαi, Gαo, Gαs and Gα12. Besides the Gαq expression in the *r-opsin*[+] photoreceptors of the eyes, the only other antibody to show specific localization was Gαi, and this was expressed in the adult phaosomal cells including on the ciliary membrane (*Figure 5I and J*). The expression of xenopsin in these cells suggests that they are photoreceptors, and this would be further supported if it could be shown that xenopsin is active and can form a photopigment. The added expression of Gαi in these cells may indicate that this is the signaling pathway of xenopsin, however many Gα subunits can be expressed in the same sensory neuron (*Kusakabe et al., 2000*). To investigate xenopsin's ability to form a photopigment and its Gα-protein coupling preference, we carried out functional assays in human HEK293 cells (see below).

## *Maritigrella* xenopsin forms a photopigment capable of sustained Gαi signalling

To determine whether *Maritigrella* xenopsin functions as a photopigment, and to explore which classes of Gα protein it can couple to, we assessed its ability to modulate levels of the second messenger molecules cyclic AMP (cAMP) and calcium ($Ca^{2+}$) in response to light when heterologously expressed in human HEK293 cells (*Bailes and Lucas, 2013*; *Koyanagi et al., 2013*). Changes in cAMP or $Ca^{2+}$ levels are characteristic of opsin coupling to the three major families of G alpha protein: Gαq, Gαs and Gαi/o/t (hereafter simply 'Gαi'). These assays were conducted with and without pertussis toxin, which selectively inactivates the Gαi family and has no effect on Gαs or Gαq, and therefore pertussis toxin sensitivity is diagnostic of Gαi family signaling. To a first approximation, Gαs coupling causes an *increase* in cAMP, and is not affected by pertussis toxin; Gαi coupling causes a *decrease* in cAMP and a small increase in cytoplasmic $Ca^{2+}$, both of these effects of Gαi signaling are abolished by pertussis toxin; and Gαq coupling causes an increase in cytoplasmic $Ca^{2+}$, but is not impacted by pertussis toxin. We compared the response of *Maritigrella* xenopsin to three positive control opsins: the cnidopsin JellyOp, human rod opsin, and human melanopsin, which are known to couple potently and selectively to Gαs, Gαi, and Gαq, respectively (*Bailes et al., 2012*; *Bailes and Lucas, 2013*). In all experiments, we used a flash of 10̇15 photons of 470 nm light as the

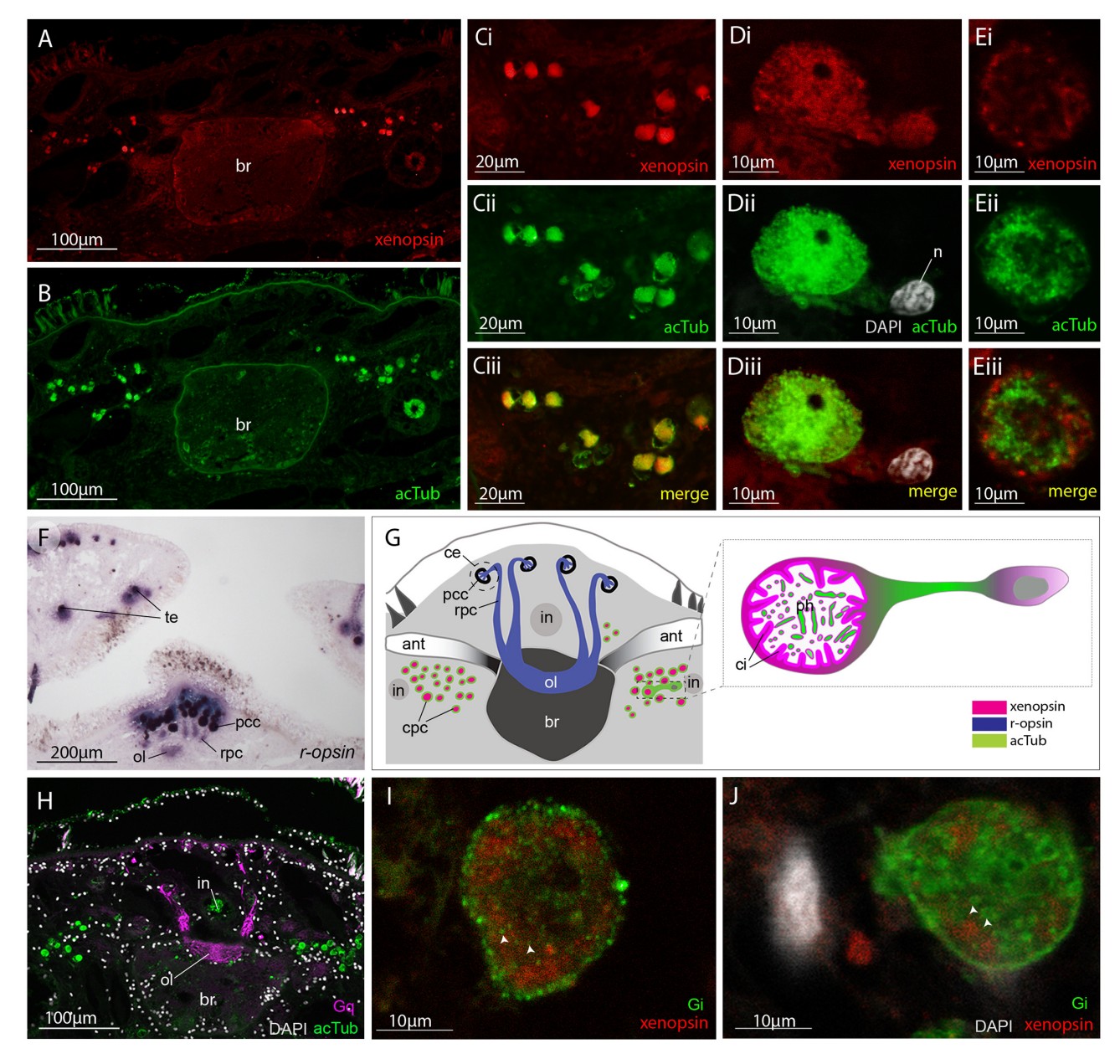

**Figure 5.** Xenopsin is co-localised with acetylated tubulin and Gαi in extraocular ciliary phaosome cells and *r-opsin* is expressed with Gαq in eyes. (A-E) Co-localization of acetylated tubulin and xenopsin is found throughout the cell (*n*, nucleus) (100% of acTub[+] cells express xenopsin (in one cross section, n = 3 individual worms). (E) A cross section of a phaosome shows stronger xenopsin expression near the base of the cilia (n = 8 individuals); (F) *r-opsin* is expressed in photoreceptor cells (rpc) that extend from the pigment cup cells (pcc) to the optic lobe (ol) of the brain, together forming the cerebral eyes (ce); *r-opsin* is also expressed in the tentacular eyes (te) (n = 5 individuals); (G) Schematic of xenopsin and *r-opsin* expression in adult cross-section, with a diagram of a putative ciliary photoreceptor cell (ciliary phaosome cell, cpc) showing co-localization of xenopsin and acetylated tubulin. (H) The position of the putative ciliary photoreceptors (labeled with acetylated tubulin) in relation to the rhabdomeric photoreceptors (labeled with Gαq) (n = 3 individuals). (I and J) Gαi expression in the xenopsin[+] cells including on the ciliary membranes (arrowheads).
DOI: https://doi.org/10.7554/eLife.45465.016

The following source data and figure supplement are available for figure 5:

**Source data 1.** Sequence data for G alpha subunits used in phylogenetic analyses for *Figure 5*.
DOI: https://doi.org/10.7554/eLife.45465.018
**Figure supplement 1.** G alpha subunit phylogeny and C terminal end alignment.
DOI: https://doi.org/10.7554/eLife.45465.017

stimulus. Second messenger levels were monitored in real time using the bioluminescent reporter proteins Glosensor cAMP 22F (Glosensor) for cAMP and Aequorin localized to the cytoplasmic surface of the mitochondria (mtAequorin) for cytoplasmic $Ca^{2+}$. GPCR-Gα specificity is nearly entirely determined by the C-terminal amino acid sequence of the Gα protein (*Flock et al., 2015*; *Flock et al., 2015*), therefore the high degree of conservation between *Maritigrella* and human Gα C-terminal ends (*Figure 5—figure supplement 1B*) suggests that human Gα proteins in HEK293 cells will interact with *Maritigrella* xenopsin similarly to *Maritigrella* Gα homologs.

All opsins were constructed with the C-terminal nine amino acids of rod opsin (TETSQVAPA) as an epitope tag for the 1D4 monoclonal antibody. Immunocytochemistry confirmed that all four opsins, including *Mc* xenopsin, were expressed in HEK293 cells at roughly similar levels, and *Mc* xenopsin fluorescent intensity was significantly greater than the no opsin control (*Figure 6* and *Figure 6—figure supplement 1*).

To assay for Gαs coupling (*Figure 6A*), cells were transfected with Glosensor and either xenopsin or JellyOp and exposed to light. As expected, JellyOp induced a > 100 fold increase over baseline in Glosensor luminescence, and no response was observed in negative control cells without opsin. JellyOp signaling was not affected by the addition of pertussis toxin, which does not interfere with Gαs. In contrast, xenopsin induced a ~ 40% decrease in Glosensor signal in response to light, suggesting Gαi coupling. The addition of pertussis toxin blocked the decrease in cAMP in xenopsin-expressing cells, confirming that it was caused by coupling to Gαi. In pertussis toxin treated cells, xenopsin drove a very small (~0.2 fold) increase in Glosensor cAMP signal, suggesting that xenopsin may also weakly couple to Gαs.

To better assay for Gαi coupling, we used the drug forskolin to artificially elevate cAMP by activating adenylyl cyclase prior to the light flash. HEK293 have low basal levels of cAMP, so forskolin treatment increases the magnitude of cAMP suppression that it is possible to achieve with a Gαi coupled opsin. Cells were transfected with Glosensor and xenopsin or rod opsin, with or without pertussis toxin pretreatment. After measuring basal Glosensor cAMP luminescence, forskolin was added, which induced a strong increase in Glosensor signal that stabilized after ~ 40 min. At this point, cells were exposed to light. Both rod opsin and xenopsin suppressed cAMP to below 50% of pre-flash levels, whereas no decrease was seen in cells without opsin. As expected, the ability of rod opsin and xenopsin to suppress cAMP was entirely blocked by pertussis toxin, confirming that cAMP suppression by xenopsin is driven by coupling to Gαi (*Figure 6B*). In pertussis toxin treated cells, the addition of forskolin amplified the cAMP-stimulating effect of xenopsin (*Figure 6B*) noted above (*Figure 6A*), providing firmer evidence for a promiscuous coupling to Gαs or another pertussis toxin-insensitive pathway, but in unperturbed cells the main effect of xenopsin was to decrease cAMP in response to light.

Although xenopsin and rod opsin both suppress cAMP by coupling to Gαi in the absence of pertussis toxin, there are intriguing differences in the kinetics of their response. Rod opsin produces a transient decrease in cAMP, which reaches a minimum at 5 min post-flash and returns to the level of control cells by 20 min. In contrast, xenopsin appears to irreversibly suppress cAMP in this system (*Figure 6B*). The long lifetime of the xenopsin response indicates that its active signaling state is very stable in this system and could also suggest that the signal termination mechanisms (e.g. GPCR kinases and beta-arrestins) present in HEK293 cells may not be compatible with xenopsin.

Finally, to investigate whether xenopsin is capable of coupling to Gαq, we tested its ability to modulate cytoplasmic $Ca^{2+}$ release, in comparison to melanopsin, with or without pertussis toxin pretreatment. Although cytoplasmic $Ca^{2+}$ release is a classic response to Gαq activation, it is known that Gαi activation can also trigger $Ca^{2+}$ release (*Mizuta et al., 2011*). In cells transfected with melanopsin, light induced a > 1000 fold increase in cytoplasmic $Ca^{2+}$, which was not affected by pertussis toxin. Xenopsin triggered a smaller ~ 100 fold increase in cytoplasmic $Ca^{2+}$, importantly, this response is abolished by pertussis toxin (*Figure 6C*). Because xenopsin did not stimulate $Ca^{2+}$ signaling in the presence of pertussis toxin, we conclude that xenopsin is not capable of coupling to Gαq. These results show that in human HEK293 cells, xenopsin forms a functional opsin that predominantly suppresses cAMP in response to light by coupling to Gαi pathways, and that it has a secondary capacity to couple to Gαs or another undefined pathway capable of elevating cAMP, but that this secondary effect is only apparent when Gαi is artificially inactivated with pertussis toxin. It is important to note that these assays were performed in non-native cells, and the effects of xenopsin

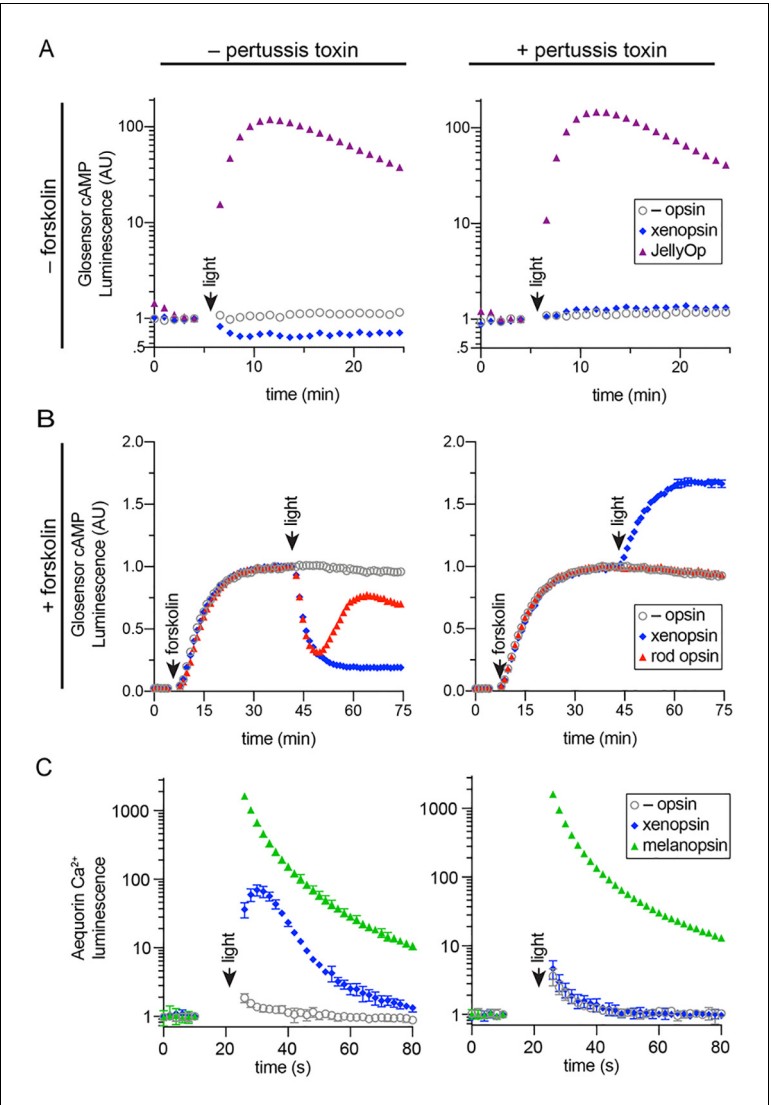

**Figure 6.** In human cells *Maritigrella crozieri* xenopsin forms a functional photopigment that predominantly couples to Gαi pathways. (**A,B**) HEK293 cells were transfected with Glo22F and indicated opsins, + /- pertussis toxin, and exposed to light. In B, cells were treated with 2 µM forskolin prior to the light flash. (**C**) HEK293 cells were transfected with mtAequorin and the opsins indicated, + /- pertussis toxin, and exposed to light. Plots show mean luminescence of technical replicates (from one representative of three biological replicates) normalized to the pre-flash timepoint, + /- SEM. Error bars smaller than symbols are not shown. n = 3 technical replicates in A,B; n = 4 technical replicates in C. The other biological replicates are shown in *Figure 6—figure supplement 2*.

DOI: https://doi.org/10.7554/eLife.45465.019

The following figure supplements are available for figure 6:

**Figure supplement 1.** Immunofluorescence to quantify opsin expression in HEK293 cells.
DOI: https://doi.org/10.7554/eLife.45465.020

**Figure supplement 2.** Two further biological replicates of the secondary messenger assays show there was quantitative variation from day to day in the magnitude of responses to light and forskolin, but the qualitative response of each opsin was consistent (excluding one replicate in which rod opsin showed no activity, possibly due to a faulty preparation of plasmid DNA).
DOI: https://doi.org/10.7554/eLife.45465.021

signaling in their native cellular context may be different, depending on the Gα proteins and other downstream factors expressed there.

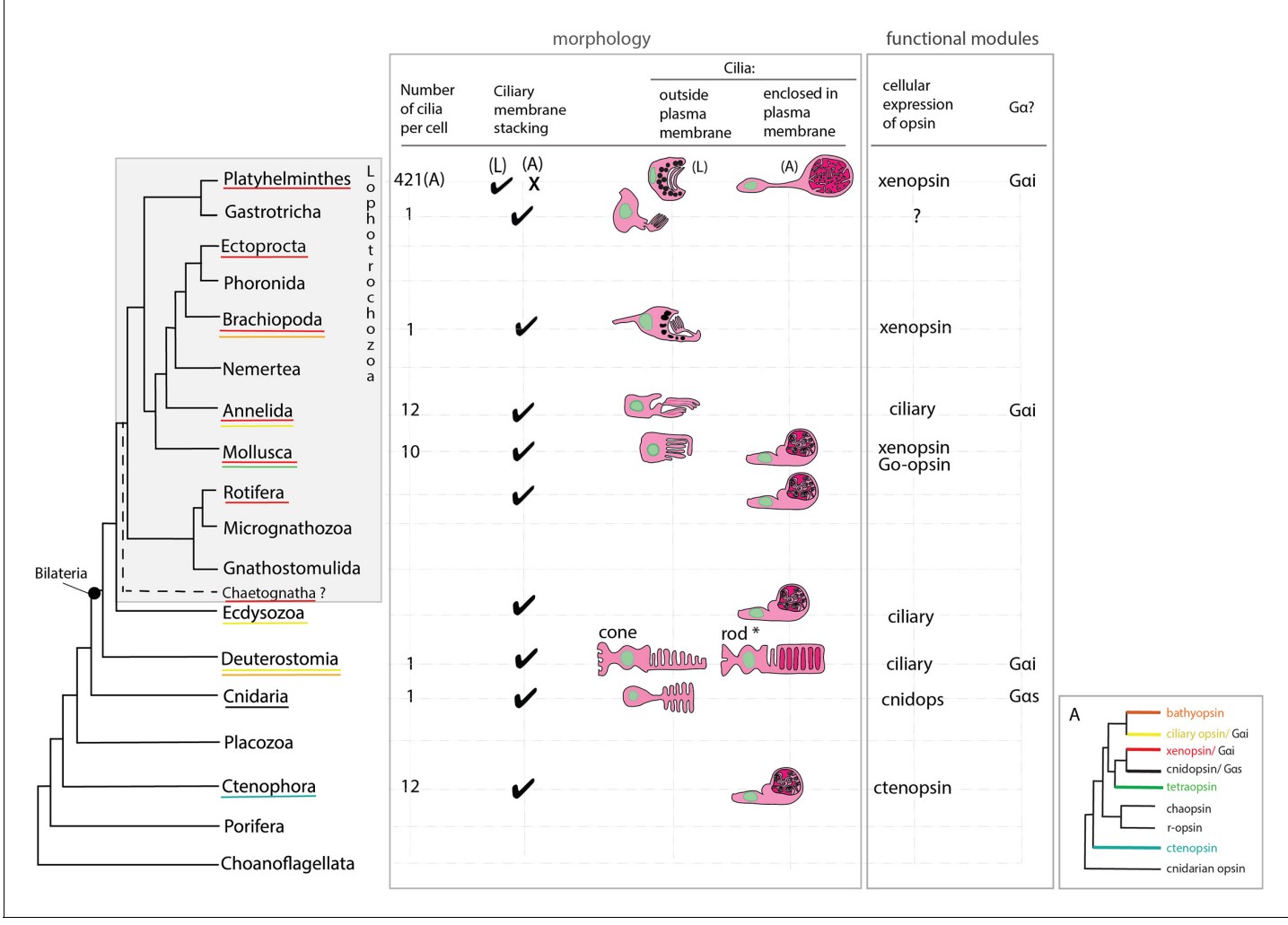

**Figure 7.** An overview of metazoan ciliary photoreceptor morphology, opsin expression and Gα-protein coupling (determined from cellular assays), highlighting the distinct morphology of the ciliary phaosomes in flatworms and possible convergent evolution of enclosed ciliary membranes in invertebrate phaosomes and jawed vertebrate rods (*). The colored lines under the phylum names represent the presence of the opsin sub-type in the taxonomic group; note the higher prevalence of xenopsins over ciliary opsins in Lophotrochozoa. (L) = larval photoreceptor, (A) = adult photoreceptor. Box A shows the opsin relationships according to our phylogeny and the known Gα-binding of opsins expressed in ciliary photoreceptors.
DOI: https://doi.org/10.7554/eLife.45465.022

## Discussion

Our phylogenetic analyzes of opsin genes across the Metazoa supports the emerging consensus that xenopsins and c-opsins diverged prior to the bilaterian common ancestor. As both opsin types are found in various protostomes, this suggests that the two opsins co-existed in the protostome stem lineage. The known taxonomic distribution of xenopsins and c-opsins is strange in this context in that no species is known in which both opsins are present. Our survey of flatworm opsins has revealed another instance in which only one of the two related xenopsin/c-opsins types is found. We find that the same is true of a chaetognath and a bryozoan. Why different groups have retained one opsin rather than the other is unknown.

One possibility to explain this distribution is that, while clearly separate clades with distinct and conserved genes structures unique to each type of opsin, the two opsins are, nevertheless, largely equivalent. In addition to their relatively close phylogenetic relationship, we have shown similarities in the shared possession of the NxQ and VxPx motifs, which, in c-opsin, allow Gαi protein binding and transport to cilia respectively. Going beyond comparative analysis of amino acid sequences, we have provided two pieces of evidence that suggest Gαi coupling of xenopsin; primary activation of a Gαi signal transduction cascade in living human cells, and xenopsin and Gαi localization in the same

putative ciliary photoreceptor in a flatworm. We have also shown xenopsin protein expression on cilia of cells that form eyes, supporting the possibility that the VxPx motif is also involved in transporting xenopsin to cilia. We have shown that conserved sequences motifs in xenopsin and c-opsin lead to similar function (Gαi binding) and expression (ciliary cells). Whether any species exist with both opsins will require broader taxonomic sampling and this may help eludicate any functional differences between them.

Our analyzes also support the idea that xenopsins themselves duplicated, as suggested by the presence of two clades of xenopsin sequences from the polyclad flatworms (*Vöcking et al., 2017*). Xenopsins from clade A have the signatures to be photopigments, while those in clade B, although they have lysine in transmembrane domain VII to bind to the chromophore, lack the NxQ and VxPx motifs. This might suggest that they are photoisomerases supporting the xenopsin photopigments of clade A by recycling the chromophore.

We have provided the first evidence that a xenopsin, *Mc* xenopsin, forms a functional photopigment. When heterologously expressed in HEK293 cells, and reconstituted with 9-*cis* retinal, *Mc* xenopsin elicits a light-dependent decrease in cAMP that is blocked by pertussis toxin, indicating that it can signal via a Gαi signal transduction cascade. *Mc* xenopsin also drove a transient increase in intracellular $Ca^{2+}$ but, as this response was blocked by pertussis toxin, it reflects crosstalk with the Gαi signaling pathway, not coupling to Gαq. Our data do indicate that *Mc* xenopsin is capable of promiscuous signaling, specifically, when Gαi is inactivated by pertussis toxin, *Mc* xenopsin acts to increase cAMP in response to light, revealing additional coupling either to Gαs or to another, undefined pertussis toxin insensitive pathway (as observed for other invertebrate opsins, such as scallop opsin 1; *Ballister et al., 2018*). Nonetheless, in the case of xenopsin, cAMP suppression through Gαi coupling is the primary effect in unperturbed HEK293 cells.

Compared to human rod opsin, *Mc* xenopsin was effectively irreversibly activated by blue light in this assay; there was no change in its signaling over tens of minutes. Although this may reflect an incompatibility with human GPCR kinases and arrestins, it could also imply that the activated state of the opsin is thermally stable, and may be bistable, with a thermally stable signal state that can be converted back to inactive dark state by subsequent light absorption (*Tsukamoto and Terakita, 2010*). Bistable opsins, such as lamprey parapinopsin, are known to exhibit prolonged cAMP suppression in response to blue light in live cell assays of Gαi activation (*Kawano-Yamashita et al., 2015*), similar to the responses observed with xenopsin. Several aspects of xenopsin signaling must be explored further. What are the relevant second messengers and signaling kinetics in its native cells, and, of the G alpha subunits we have identified, which ones, other than Gαi, are expressed there? What are the spectral sensitivity, quantum efficiency, and cofactor requirements of xenopsin? These questions could be addressed by a combination of in vivo electrophysiology, in vitro secondary messenger assays directly in cultured phaosome cells and single cell RNA sequencing.

Xenopsin's ability to drive phototransduction in live human cell assays and its expression in ciliary cells in *Maritigrella* add molecular and functional evidence (albeit indirect) for the presence of ciliary photoreceptors in flatworms. Metazoan ciliary photoreceptors are morphologically diverse, especially in the outer segment where the ciliary membranes increase cell surface area through membrane stacking or greater numbers of cilia, or both. Some CPRs project their ciliary membranes into the extra-cellular environment, whereas others enclose these membranes within their own plasma membranes; and some CPRs are supported by shading pigment forming an eye whereas others are extra-ocular. In *Maritigrella*, xenopsin was expressed in three types of ciliary cell each with distinct ciliary arrangements; the modified cilia of the larval epidermal eye, the unmodified cilia of the cerebral eye in the larva (which project into extra-cellular space), and the unmodified cilia enclosed in phaosomes in extraocular cells in the adult. Xenopsin expression in the ciliary cells of the larval eyes is similar to expression in other larval Lophotrochozoa; the purely ciliary eye of the brachiopod (*Passamaneck et al., 2011*), and the mixed ciliary/microvillar, *xenopsin* and *r-opsin* expressing eye in a chiton (*Vöcking et al., 2017*); except that in the chiton *xenopsin* and *r-opsin* are co-expressed in a cell that bears both cilia and microvilli, whereas in polyclad larvae the cerebral eye consists of distinct rhabdomeric and ciliary cells (*Eakin and Brandenburger, 1981*).

The unusual morphology, and the extraocular and extra-cerebral location of the xenopsin[+] cells in the adult makes them distinct from ciliary photoreceptors in other Metazoa, and only by demonstrating a photoresponse and phototransduction in these cells in the live worms would we have direct evidence that they are ciliary photoreceptors. With this caveat in mind, however, we can compare

them to ciliary photoreceptors in other animals. Their unique morphology led to the idea that they are a flatworm novelty (*Sopott-Ehlers et al., 2001*), and two lines of evidence support this idea (*Figure 7*). Firstly, while most metazoan CPRs project their ciliary membranes outside of the cell (like jawed vertebrate cones), *Maritigrella* ciliary phaosomes are more 'rod-like' because they enclose their ciliary membranes within their own plasma membrane, like gnathostome rods and a few examples of ciliary phaosomes from other invertebrates (*Figure 7*). Secondly, *Maritigrella* phaosome cells increase their surface area by increasing the number of cilia rather than modifying the ciliary membranes into discs or lamellae (*Figure 7*). The sheer number of cilia housed within the phaosome is striking and may be a unique feature of *Maritigrella* and other flatworm CPRs. A few examples of unmodified cilia in complete or open phaosomes have been recorded in other lophotrochozoans (*Hessling and Purschke, 2000*; *Woollacott and Zimmer, 1972*), but most other phaosomal CPRs have modified cilia (flattened and whorled) (*Clement and Wurdak, 1984*; *Boyle, 1969*; *Horridge, 1964*).

The functional benefits of enclosing the ciliary membranes are not known, even in rods, but suggestions include increased efficiency in the transport of photopigments, the renewal of the outer segments, and of separating ion channels on the plasma membrane from opsins and other transduction proteins on the ciliary membrane (*Morshedian and Fain, 2015*; *Morshedian and Fain, 2017*). The patchy phylogenetic distribution of cells that enclose their ciliary processes suggests this trait has evolved independently in these taxa (*Figure 7*). Phaosomes are not specific to ciliary photoreceptors, in the Clitellata, a major annelid group, microvillar, *r-opsin*-expressing phaosomes are the only type of photoreceptor found (*Döring et al., 2013*).

These studies on invertebrate phaosomes will facilitate comparative studies with gnathostome rods to understand the function and evolution of enclosed photosensitive membranes. Rods evolved from cones (*Kawamura and Tachibanaki, 2008*; *Morshedian and Fain, 2015*) to give highly sensitive but coarse monochromatic scotopic vision at night. Physiologically, the defining feature of a rod is its ability to respond to single photons of light in the dark-adapted state (*Baylor et al., 1979*). This high sensitivity of rods is further enhanced by very slow response kinetics, slow transduction machinery and a slow recovery of visual sensitivity following bleaching, compared to cones (summarized in Warrant 2015). Whether any of these properties are seen in xenopsin in its native cells will require further investigation, but in human cells at least *Mc* xenopsin does have a prolonged and sustained response to light.

Our morphological analyzes show that the phaosomal cells of *Maritigrella* adults are not associated with pigment cup cells, they are therefore extraocular and could detect light from all directions. Animals use non-directional light to set circadian cycles, monitor UV levels and photoperiodism, to gauge depth, and to detect a predator's shadow (*Nilsson, 2009*). Cells that only signal slow changes of ambient light intensity, however, can work without membrane specializations (*Nilsson, 2004*), such as intrinsically photosensitive retinal ganglion cells (*Hattar et al., 2002*). The density of phaosome cells in *Maritigrella*, and the number of cilia in each phaosome, increases the surface area to allow higher concentrations of photopigment, which would suggest very high sensitivity to light. The ciliary surface area of a phaosome is approximately four times smaller than the total disc membrane surface of rat rods (*Mayhew and Astle, 1997*) but two times larger than that of the brain CPRs of the annelid *Platynereis* (*Verasztó et al., 2018*). High sensitivity is normally associated with a visual role (*Nilsson, 2004*) but as all extraocular photoreceptors are, by definition, non-visual (*Cronin and Johnsen, 2016*) then their membrane elaborations could indicate that these photoreceptors function to detect intermediate to fast changes of non-directional light associated with changes in depth or a predator's shadow, rather than (or as well as) the slower changes of light over 24 hr or seasons. If *Mc* xenopsin in its native cell shows the same sustained response that we show here in mammalian cells, then perhaps these cells function to detect and amplify low levels of light, like vertebrate rods, and could be involved in detecting moonlight.

The adult xenopsin+ ciliary phaosomes of *Maritigrella* are extraocular like the CPRs of the marine annelid *Platynereis.* In the annelid these are located in the brain and express a UV-sensitive ciliary opsin (*Arendt et al., 2004*; *Tsukamoto et al., 2017*). These cells control circadian behaviors via melatonin production (*Tosches et al., 2014*) and mediate downward swimming in response to non-directional UV light in the larval stage (*Verasztó et al., 2018*). The *Platynereis* c-opsin binds to exogenous all-*trans*-retinal, which is particularly important for opsins expressed outside of the eyes (*Tsukamoto et al., 2017*) where sophisticated multi-enzyme systems producing a thermally unstable

11-*cis*-retinal isomer probably don't exist (*Yau and Hardie, 2009*). Although *Mc* xenopsin was able form a photopigment with 9-*cis*-retinal it would be interesting to examine retinal isomer preferences.

The discovery of *xenopsin*⁺ ciliary cells in the eyes of protostome larvae challenged our views on opsin and photoreceptor evolution by deviating from the invertebrate trend of extraocular ciliary photoreceptors and ocular rhabdomeric/microvillar photoreceptors (*Passamaneck et al., 2011*; *Vöcking et al., 2017*). Xenopsin expression is both ocular and extraocular, however, either simultaneously (e.g. in the chiton larva [*Vöcking et al., 2017*]) or sequentially over development as we have shown in *Maritigrella* (e.g. in the larval eyes and adult extraocular ciliary phaosomes), and this is a feature of most opsin types (*Porter et al., 2012*; *Cronin and Porter, 2014*; *Sprecher and Desplan, 2008*). Our findings show that xenopsin is localized in different types of cillary cell, with different functions (ocular and extraocular) at different points in development, highlighting the importance of including developmental data into evolutionary scenarios on photoreceptor evolution. In fact, xenopsin in the extra-ocular ciliary phaosomes of adult *Maritigrella* is yet another similarity with ciliary opsin (which is expressed in extra-ocular CPRs of annelids and arthropods) and is consistent with the invertebrate trend of extra-ocular ciliary photoreceptors.

In *Maritigrella*, it is puzzling that the ciliary phaosomes in the epidermis of the larva do not express xenopsin, whilst the ciliary phaosomes in the adult do. Ultrastructurally, these cells look similar in both stages, so it is possible that the larval cells migrate sub-epidermally over ontogeny. Perhaps in the larval stage they are not fully developed, or they express another type of opsin. Marine animals such as *Maritigrella crozieri* could use light in many ways. In addition to light as a visual stimulus, light intensity and wavelength can provide clues as to the time of day, the season, the state of the tide and water depth. Light will also be used differently by a minute swimming larva and a large crawling adult worm. We have characterized a new type of opsin-expressing cell in a flatworm and demonstrated that a recently classified opsin is capable of photosensitivity and phototransduction. This adds to the increasing diversity of animal photoreceptors and phototransduction pathways being discovered as more species are studied.

# Materials and methods

## Key resources table

| Reagent type (species) or resource | Designation | Source or reference | Identifiers | Additional information |
|---|---|---|---|---|
| Antibody | anti-acetylated tubulin (monoclonal, mouse) | Sigma | T7451 RRID:AB_609894 | IF(1:500) |
| Antibody | Gα q/11 (C-19) (polyclonal, rabbit) | Santa Cruz Biotech | sc-392 RRID:AB_631537 | IF(1:300) |
| Antibody | Gαi-1 (R4) (monoclonal, mouse) | Santa Cruz Biotech | sc-13533 RRID:AB_2111358 | IF(1:300) |
| Antibody | Gα$_i$-1/2/3 (35) (monoclonal, mouse) | Santa Cruz Biotech | sc-136478 RRID:AB_2722559 | IF(1:300) |
| Antibody | Gα$_o$ (A2) (monoclonal, mouse) | Santa Cruz Biotech | sc-13532 RRID:AB_2111645 | IF(1:300) |
| Antibody | Gα$_{s/olf}$ (A-5) (monoclonal, mouse) | Santa Cruz Biotech | sc-55545 RRID:AB_831819 | IF(1:300) |
| Antibody | Gα$_{s/olf}$ (C-18) (polyclonal, rabbit) | Santa Cruz Biotech | sc-383 RRID:AB_631539 | IF(1:300) |
| Antibody | Gα12 (E-12) (monoclonal, mouse) | Santa Cruz Biotech | sc-515445 | IF(1:300) |
| Cell line (*Homo-sapiens*) | Embryonic kidney cells | ATCC | CRL3216 RRID:CVCL_0063 | |

*Continued on next page*

*Continued*

| Reagent type (species) or resource | Designation | Source or reference | Identifiers | Additional information |
|---|---|---|---|---|
| Recombinant DNA reagent | pGlosensor 22 | Promega | E1290 | Live-Cell Biosensors |
| Recombinant DNA reagent | pcDNA3.1 | Invitrogen | V79020 | Mammalian Expression Vector |
| Recombinant DNA reagent | pcDNAFRT/TO vector | Thermo | V652020 | Expression vector |
| Chemical compound | 9-*cis* retinal | Sigma | R5754 | |

## Identification of opsin and Gα subunit sequences

*Maritigrella crozieri* ciliary-type and rhabdomeric opsins, and Gα subunits were identified by reciprocal best match BLAST searches on a mixed stage embryonic and larval transcriptome (*Lapraz et al., 2013*). Four *Schmidtea mediterranea* opsin sequences (*Zamanian et al., 2011*) – Additional File 7] along with the *Maritigrella* sequences were used, with Mollusc opsins (*Ramirez et al., 2016*), as query sequences for BLAST searches against assembled transcriptomes and genomes for 30 other flatworm species (*Egger et al., 2015*; *Laumer et al., 2015*), a chaetognath (*Pterosagitta draco*) and a bryozoan (*Bugula neritina*). Our search allowed the identification of some already published flatworm sequences (*Vöcking et al., 2017*) (DAB27256.1, DAB27257.1, DAB27258.1, DAB27259.1, DAB27253.1, DAB27254.1 and DAB27255.1). These published accession numbers and sequences were used in our analysis. We used Gα subunit proteins from *Terebratalia transversa* (*Passamaneck et al., 2011*) and *Platynereis dumerilii* as blast query sequences against the *Maritigrella* transcriptome and the genome of another flatworm *Schistosoma mansoni* (www.parasite.wormbase.org).

## Phylogenetic analysis

Flatworm, bryozoan and chaetognath best hit opsin sequences were added to a subset of the *Ramirez et al. (2016)* metazoan opsin sequences dataset. The subset was obtained by first reducing redundancy of the original dataset using an 80% identity threshold with CD-HIT (*Li and Godzik, 2006*), then by discarding sequences which, in an alignment, did not fully cover the region found between the first and last transmembrane regions. Finally, when multiple sequences belonging to the same taxonomic clade or class were found, only the two or three most complete representative sequences where kept. Additional opsin sequences were added to the dataset: human melanopsin, human rhodopsin and *Carybdea rastonii* opsin (called JellyOp in this study), *Xenopus laevis* OPN4B (covering a taxonomic gap), *Acromegalomma interruptum* InvC-opsin and *Spirobranchus corniculatus* InvC-opsin (kindly provided by Dr. Michael Bok, *Bok et al., 2017*), *Owenia_fusiformis* Xenopsin1, 2 and 3 (*Vöcking et al., 2017*), *Platynereis dumerilii* TMT1 (http://genomewiki.ucsc.edu/index.php/Opsin_evolution), as well as additional non-opsin outgroup sequences (Prostaglandin E2 receptor and Melatonin receptor sequences) resulting in a first dataset (Dataset 1) of 213 sequences. In order to evaluate their influence on the tree topology, sequences forming small monophyletic groups (Bathyopsin, Chaopsin, ctenophore and cnidarian early branching opsins in Ramirez et al. opsin phylogeny [*Ramirez et al., 2016*]) were removed from our initial dataset (Dataset 2–196 sequences).

For both datasets, sequences were aligned with MAFFT (*Katoh and Standley, 2013*) webserver (https://mafft.cbrc.jp/alignment/server/) using the L-INS-I option. Portions of the alignment with fewer than six represented positions were trimmed from the alignment using trimAl v1.2rev57 (*Capella-Gutiérrez et al., 2009*), then the alignment was manually trimmed to remove positions before first aligned methionine and after the last aligned block.

For both datasets, Maximum-Likelihood phylogenetic reconstruction of the trimmed alignment was conducted using both: IQ-TREE webserver (http://iqtree.cibiv.univie.ac.at/) (*Trifinopoulos et al., 2016*) with a LG+R9+F substitution model, and with 1000 Ultrafast bootstrap replications as well as SH-aLRT (1000 replicates) and approximate aBayes single Branch testing, or with RAxML v.8.2.9

(*Stamatakis, 2014*) on the Cipres webserver (www.phylo.org/portal2/) (*Miller et al., 2010*) with a GAMMA-LG-F substitution model and 100 rapid boostrap replicates.

*Maritigrella crozieri* best hit Gα subunits sequences were added to Gα subunit sequences from multiple taxa obtained from *Passamaneck et al. (2011)*. Additional Gα subunit sequences were added to the dataset: *Lymnea stagnalis* Gαi, Gαo and Gαs, *Platynereis dumerilii* Gαi and Gαq, *Schistosoma mansoni* Gαi1, Gαi2, Gαi3, Gαo, Gαq1, Gαq2, Gαs1 and Gαs2 and *Arabidopsis thaliana* GPA1. Sequences were aligned and alignment trimmed using the same methods and parameters used for opsin sequences and alignment. Maximum-Likelihood phylogenetic reconstruction of the trimmed alignment was conducted using IQ-TREE webserver (http://iqtree.cibiv.univie.ac.at/) (*Trifinopoulos et al., 2016*) with a LG+I+G4 substitution model, and with 1000 Ultrafast bootstrap replications as well as SH-aLRT (1000 replicates) and approximate aBayes single Branch testing.

FigTree v1.4.3 (tree.bio.ed.ac.uk/software/figtree/) was used for tree visualization. Accession numbers of the sequences used in the phylogenetic analysis are available in the *Figure 1—source data 1*.

## The morphology, and opsin expression, of *Maritigrella crozieri* ciliary photoreceptors

### Animal collection, fixation and sectioning

Adult *Maritigrella crozieri* were collected from the Florida Keys (*Rawlinson, 2010*; *Lapraz et al., 2013*). They were fixed in a Petri-dish containing frozen 4% paraformaldehyde (diluted in phosphate buffered saline [PBS]) overnight at 4°C, rinsed in PBS (3 × 5 min, 5 × 1 hr washes) at room temperature and dehydrated in a step-wise ethanol series for histology and immunofluorescence, and in a methanol series for mRNA in situ hybridization. For histology, heads of adult worms (from the pharynx anteriorward) were dissected, cleared in histosol (National Diagnostics), and embedded in paraffin. Paraffin blocks were sectioned at 8–12 µm using a Leica (RM2125 RTF) microtome. Larval stages were fixed in 4% PFA in PBS for 20 min at room temperature, rinsed in PBS for five x 30 min washes and stored in 1% PBS-azide at 4°C for immunofluorescence, or dehydrated into 100% methanol and stored at −20°C for mRNA in situ hybridization.

### Histology and immunohistochemistry

For adult stages, consecutive sections were used to compare histology and immunofluorescence. For histological analysis, sections were stained with Masson's trichrome (MTC) (*Witten and Hall, 2003*). For immunostaining of paraffin sections, slides were dewaxed in Histosol (2 × 5 min), then rehydrated through a descending ethanol series into PBS + 0.1% Triton (PBT, 2 × 5 min). An antigen retrieval step, heating on low in a microwave in sodium citrate buffer (10 mM sodium citrate, pH6.0) for 1 min, helped with the xenopsin and Ga subunit primary antibodies. The slides were rinsed 2 × 5 min in PBS + 0.1% triton, before blocking with 10% heat-inactivated sheep serum in PBT for 1 hr at room temperature in a humidified chamber. Primary antibodies (see below) were diluted in block (10% heat-inactivated sheep serum IN PBT) and applied to the slide, covered with parafilm, and incubated at 4°C for 48 hr. Slides were then rinsed in PBT (3 × 10 min). Secondary antibodies diluted in block solution were then applied to each slide, and slides were covered with parafilm and incubated in a humidified chamber, in the dark, at room temperature for 2 hr. Slides were rinsed in PBT 3 × 10 min, and then 4 × 1 hr prior to counterstaining with the nuclear marker 4′, 6-diamidino-2-phenylindole (DAPI) (1 ng/ml) and mounting in Fluoromount G (Southern Biotech, Birmingham, AL). Immunostaining of larval stages was performed according to *Rawlinson (2010)*.

Primary antibodies used were: anti-acetylated tubulin (Sigma) diluted at 1:500, a polyclonal antibody directed against the C-terminal extremity of the *Maritigrella* xenopsin protein sequence (GASAVSPQNGEESC; generated by Genscript, Piscataway, NJ, USA) diluted at 1:50, and commercially available antibodies against Gαq/11α (C-19), Gαi-1 (R4), Gαi-1/2/3 (35), Gαo (A2), Gαs/olf (A-5), Gαs/olf (C-18), Gα12 (E-12) diluted at 1:300 (Santa Cruz Biotechnology, Santa Cruz, CA, USA). Imaging of immunofluorescence on paraffin sections and larval wholemounts was carried out using an epi-fluorescence microscope and a confocal laser scanning microscope, additional images on larvae were taken with an OpenSPIM (*Girstmair et al., 2016*). For the 3D rendering of the larva a multi-view stack was produced by capturing several angles of the specimen and using Fiji's bead based registration software and multi-view deconvolution plugins (*Preibisch et al., 2010*; *Preibisch et al., 2014*).

## mRNA in situ hybridization

To analyze the expression of *Maritigrella crozieri r-opsin*, we performed mRNA in situ hybridization using a riboprobe generated against the *r-opsin* sequence identified above. A 523 bp fragment of *M. crozieri r-opsin* was PCR amplified using the following primers: *r-opsin*-fw TCCCTGTCC TTTTCGCCAAA, *r-opsin*-rv TATTACAACGGCCCCCAACC. The fragment was cloned using the pGEM-T easy vector system, and a DIG-labeled antisense probe was transcribed according to the manufacturer's protocol. mRNA in situ hybridization on paraffin sections of adult tissue was carried out according to *O'Neill et al. (2007)*. Upon completion of the color reaction, slides were cover-slipped with Fluoromount G. Wholemount mRNA in situ hybridization on larvae was carried out according to the *Capitella teleta* protocol of *Seaver and Kaneshige (2006)*. Following termination of the color reaction, specimens were cleared and stored in 80% glycerol, 20% 5 × PBS. Both adult and larval mRNA in situ hybridization experiments were imaged on a Zeiss Axioscope.

## TEM and serial SEM

Larval *Maritigrella crozieri* were fixed in 3% gluteraldehyde in seawater overnight at 4°C. Then stored at 4°C in seawater plus 0.1% glutaraldehyde.

Adult *Maritigrella crozieri* heads were dissected and immediately placed in ice-cold, freshly prepared 3% glutaraldehyde overnight at 4°C. The tissue was rinsed seven times in 0.1M sodium phosphate buffer, pH 7.2, then placed in 1% osmium tetroxide (in the same buffer) for 1 hr at 4°C. Samples were then rinsed twice with ice-cold distilled water and dehydrated in an ethanol series (50%, 75%, once each for 15 min; 95%, 100% twice each for 15 min), culminating in two changes of propylene oxide with a waiting period of 15 min after each change. The samples were then placed in Epon mixture/propylene oxide (1:1) for 45 min at room temperature (22–25°C). Finally, samples were transferred from vials into fresh Epon mixture in molds and polymerized in an oven at 60°C for 72 hr.

For TEM, sections of 60–70 nm thickness were cut with a diamond knife on a Reichert Ultracut E ultramicrotome. After their collection on formvar film coated mesh grids, the sections were counter-stained with lead citrate. The ultrathin sections were analyzed using a Jeol-1010 electron microscope at 80 kV mounted with a Gatan Orius camera system.

For serial SEM the samples were shaped to an ~1×4 mm rectangular face using a diamond trimming tool. The block was mounted in a microtome (Leica EM UC7, Buffalo Grove, IL) and thin sections, 100, 400 and 500 nm in thickness, were cut with a diamond knife. The methods are described in detail in *Terasaki et al. (2013)* but in brief the sections were collected on kapton tape with the ATUM tape collection device, the tape containing the sections was cut into strips, mounted on four inch silicon wafers and then carbon coated. The sections were imaged using a field emission scanning EM (Zeiss Sigma FE-SEM, Peabody) in backscatter mode (10 keV electrons,~5 nA beam current). The images were aligned using the Linear Alignment with SIFT algorithm and reconstructed using TrakEM2, both in FIJI Image J (*Cardona et al., 2012*). To estimate the sensory membrane surface area of the phaosomal cells we counted the number of basal bodies (in two complete phaosomes) and calculated the average diameter and total length of 3 cilia per phaosome.

## Observations of ciliary phaosomes in live adult worms

To investigate whether the cilia in the phaosomes of live adult worms were motile, we gently squeezed small adults in seawater between a coverslip and microscope slide and observed them under a dissecting microscope, while changing the levels of illumination using a SCHOTT AG Lighting and Imaging KL 1600 LED Cold Light Source.

## Micro-CT analysis

One adult *Maritigrella crozieri* was fixed in 4% PFA, rinsed in PBS and dehydrated into methanol, as described above. It was then stained in 1% (w/v) phosphotungstic acid (Sigma 221856) in methanol for 7 days, with the solution changed every other day. The animal was rinsed in methanol, mounted in an eppendorf tube between two pads of methanol-soaked tissue paper, and scanned on a Nikon XTH225 ST at the Cambridge Biotomography Centre (Department of Zoology, University of Cambridge). The brain area was segmented using Mimics software (Materialise, Leuven, Belgium).

## Gα-protein selection of *Maritigrella crozieri* xenopsin

We followed the methods for the secondary messenger assays as described in detail in *Bailes and Lucas (2013)*. In brief, a mammalian expression vector was constructed using pcDNA3.1 (Invitrogen) and the open reading frame of *Maritigrella* xenopsin with the stop codon replaced by a six base linker and 28 bases that code for the 1D4 epitope from bovine Rh1 opsin. Expression vectors for the positive controls (Gαs – Jellyfish opsin [JellyOp]; Gαi – human rhodopsin [Rh1]; Gαq – human melanopsin [Opn4]) were constructed in the same way (*Bailes and Lucas, 2013*). Opsin-expressing plasmids were omitted from transfection in the negative controls. To make an expression plasmid for a luminescent cAMP reporter, the region for the Glosensor cAMP biosensor was excised from pGlosensor 22 (Promega) and ligated into linearized pcDNA5/FRT/TO. All restriction enzymes were from New England Biolabs (NEB). A luminescent calcium reporter was synthesized using the photoprotein aequorin from *Aequorea victoria* mtAeq (*Inouye et al., 1985*; *Bailes and Lucas, 2013*).

## Reporter and opsin transfection for light response assays

~$6 \times 10^4$ HEK293 cells (ATCC [STR authenticated], and negative for mycoplasma) were plated per well in a 96 well plate 24 hr prior to transfection in DMEM/10% FCS. Transfections were carried out using Lipofectamine 2000 (Invitrogen). Reporter and opsin-expressing plasmids were co-transfected at 500 ng each and incubated for 4–6 hr at 37°C. DMEM/10% FCS + 10 µM 9-*cis* retinal (Sigma) was then replaced and cells were left overnight at 37°C. All steps following initial transfection were carried out in dim red light only.

## Luminescent second messenger assays

We tested three biological replicates per treatment, with each biological replicate consisting of an average of three technical replicates (for cAMP assays) or four technical replicates (for Ca2+ assays).

## cAMP increases: Gαs

For measurements of cAMP increases as an indication of Gαs activity, wells of cells were transfected with pcDNA/FRT/TO Glo22F and opsin. Following transfection and overnight incubation, media was replaced with L-15 medium, without phenol red (Invitrogen), 10% FCS with 2 mM beetle luciferin (Promega) for 1–2 hr at room temperature. Luminescence of the cells was measured with a Fluostar optima plate reader (BMG Labtech). After 6 min, cells were exposed to a flash of 470 nm light ($10^{15}$ photons) followed by a recovery period where relative luminescence units (RLU) were recorded every minute for up to 25 min.

## cAMP decreases: Gαi

Decreases in cAMP are difficult to measure from baseline cAMP reporter luminescence and so cells were treated with 2 µM forskolin to artificially raise cAMP levels at 6 min. Luminescence was measured before and after the forskolin addition until the increase in luminescence plateaued. Cells were then flashed with 470 nm light (as above) and luminescence measured for up to 45mins.

## Ca2+ increases: Gαq/11

Cells transfected with pcDNA5/FRT/TO mtAeq and opsins were incubated with 10 µM Coelenterazine *h* (Biotium) in L-15 medium, without phenol red (Invitrogen), 10%FCS in the dark for 2 hr before recording luminescence on the plate reader. After 10 s, cells were flashed with 470 nm light ($10^{15}$ photons) before immediately resuming recording for 60 s.

## Immunocytochemistry of opsin expressing HEK293 cells

For immunocytochemistry, HEK293 cells were seeded in 12-well plates at 250 000 cells/well and transiently transfected, as described above. 4–6 hr later the total volume of resuspended cells was then seeded onto poly-D-lysine coated 12 mm #1.5 coverslips in each well of a 6-well plate. Cells were incubated for 24 hr at 37°C, washed in 1 x PBS and then fixed using 4% paraformaldehyde in PBS for 10 min at room temperature. Cells were washed in PBS, then permeabilised in 0.2% triton-X in PBS for 5 min. Cells were blocked in PBS with 5% goat serum and 0.05% Tween-20 for 30 mins at room temperature.

Cells were incubated in 1:500 dilution of monoclonal 1D4 rod opsin antibody (Abcam, catalog no. ab5417, lot no. GR272982-11, RRID AB_304874) in PBS with 1% goat serum and 0.05% Tween-20 for 1 hr at room temperature, then washed with PBS + 0.05% Tween-20 and incubated in 1:500 dilution of goat anti-mouse Alexa555 secondary antibody (Molecular Probes, catalog no. # A-21424) in PBS +0.05% Tween-20 with 1% goat serum for 30 min at room temperature in the dark. The secondary antibody was then removed and cells were washed three times in PBS+0.05% Tween-20. Coverslips with stained cells were mounted on slides using Prolong Gold anti-fade media with DAPI and allowed to dry at room temperature in dark for at least 24 hr.

Images were acquired using a Zeiss AxioObserver inverted microscope with a 20x/0.5 Plan-Neofluar objective and an Axiocam MRm Rev.3 camera. Ten randomly selected fields were imaged per condition. Images were analyzed using ImageJ (*Schneider et al., 2012*). Relative levels of fluorescence intensity were quantified by measuring the integrated intensity of each field above a threshold (10), then normalizing this to the average integrated intensity of the negative control (no opsin). ANOVA was carried out to compare the fluorescence intensity between the no opsin control and each opsin in turn. Representative images of opsin expression in HEK293 cells were taken with a x63 objective.

## Acknowledgements

We thank Andrew Gillis and Ariane Dimitris for help in the field, Kasia Hammar, Paul Linser, Anne Zakrzewski and Sidney Tamm for support in electron microscopy fixation, analysis and interpretation, Elaine Seaver and Danielle de Jong for helping to establish an in situ hybridization protocol for the *Maritigrella crozieri* larval stage, and Matthew Berriman and Michael Akam (Isaac Newton Trust grant (15.4(n)) for support in securing funding to complete this work. We thank the two reviewers whose comments and suggestions helped improve and clarify this manuscript.

The research was supported by a a Wellcome Trust Janet Thornton Fellowship (WT206194) to KR, a research exchange award to KR through the EDEN NSF Network (National Foundation Grant Number IOS-0955517 to Cassandra G Extavour), a Natural Sciences and Engineering Research grant (A5056) to BKH and by a Biotechnology and Biological Sciences Research Council grant (BB/H006966/1) (FL) and a Leverhulme Trust grant (F/07 134/DA) to MT. MT is supported by European Research Council (ERC-2012-AdG 322790).

## Additional information

### Funding

| Funder | Grant reference number | Author |
| --- | --- | --- |
| National Science Foundation | EDEN research exchange grant | Kate A Rawlinson |
| Wellcome | WT206194 | Kate A Rawlinson |
| Biotechnology and Biological Sciences Research Council | BB/H006966/1 | Maximilian J Telford |
| Leverhulme Trust | F/07 134/DA | Maximilian J Telford |
| H2020 European Research Council | ERC-2012-AdG 322790 | Maximilian J Telford |
| Natural Sciences and Engineering Research Council of Canada | A5056 | Brian Hall |

The funders had no role in study design, data collection and interpretation, or the decision to submit the work for publication.

### Author contributions

Kate A Rawlinson, Conceptualization, Data curation, Formal analysis, Funding acquisition, Validation, Investigation, Visualization, Methodology, Writing—original draft, Project administration, Writing—

review and editing; Francois Lapraz, Formal analysis, Investigation, Visualization, Methodology, Writing—original draft, Writing—review and editing, Phylogenetic analysis of opsins and G alpha subunits; Edward R Ballister, Formal analysis, Supervision, Validation, Investigation, Visualization, Methodology, Writing—original draft, Writing—review and editing, Xenopsin cellular assays and immunocytochemistry analysis; Mark Terasaki, Investigation, Visualization, Methodology, Serial SEM sectioning, imaging and alignment; Jessica Rodgers, Supervision, Investigation, Writing—review and editing, Xenopsin cellular assays and opsin immunocytochemistry fluorescence experiment; Richard J McDowell, Investigation, Xenopsin cellular assays and opsin immunocytochemistry experiment; Johannes Girstmair, Formal analysis, Investigation, Visualization, Methodology, Writing—original draft, Writing—review and editing, OpenSpim microscopy images and initial xenopsin immunofluorescence experiments in larvae; Katharine E Criswell, Formal analysis, Investigation, Visualization, Micro-CT reconstruction of adult brain; Miklos Boldogkoi, Investigation, Xenopsin cellular assays; Fraser Simpson, Investigation, Visualization; David Goulding, Investigation, Visualization, Methodology, Embedding of larvae for TEM analysis of ciliary photoreceptors, sectioning and imaging; Claire Cormie, Investigation, Visualization, Methodology, TEM analysis of ciliary photoreceptors in larvae, sectioning and imaging; Brian Hall, Funding acquisition, Writing—review and editing; Robert J Lucas, Resources, Supervision, Funding acquisition, Writing—review and editing; Maximilian J Telford, Resources, Supervision, Funding acquisition, Writing—original draft, Writing—review and editing

#### Author ORCIDs
Kate A Rawlinson (iD) https://orcid.org/0000-0001-8297-8405
Francois Lapraz (iD) http://orcid.org/0000-0001-9209-2018
Edward R Ballister (iD) http://orcid.org/0000-0002-4478-2513
Mark Terasaki (iD) https://orcid.org/0000-0003-4964-9401
Richard J McDowell (iD) http://orcid.org/0000-0002-6051-106X
Johannes Girstmair (iD) http://orcid.org/0000-0001-9029-3625
Katharine E Criswell (iD) https://orcid.org/0000-0002-4004-0192
Robert J Lucas (iD) http://orcid.org/0000-0002-1088-8029
Maximilian J Telford (iD) http://orcid.org/0000-0002-3749-5620

#### Decision letter and Author response
Decision letter https://doi.org/10.7554/eLife.45465.047
Author response https://doi.org/10.7554/eLife.45465.048

## Additional files

#### Supplementary files
• Transparent reporting form DOI: https://doi.org/10.7554/eLife.45465.023

#### Data availability
Nucleotide sequences for *Maritigrella crozieri* xenopsin and r-opsin have been deposited in GenBank (MK598846 and MK598847). Eight more opsin sequences from different species were identified in our study and can be found in the source data 1 for Figure 1.

The following datasets were generated:

| Author(s) | Year | Dataset title | Dataset URL | Database and Identifier |
|---|---|---|---|---|
| Rawlinson KA, Lapraz F, Ballister ER, Terasaki M, Rodgers J, McDowell RJ, Johannes Girstmair J, Criswell KE, Boldogkoi M, Simpson F, Goulding D, Cormie C, Hall BK, Lucas RJ, | 2019 | Nucleotide sequences for *Maritigrella crozieri* xenopsin | https://www.ncbi.nlm.nih.gov/nuccore/MK598846 | GenBank, MK598846 |

Telford MJ

| | | | | | |
|---|---|---|---|---|---|
| Rawlinson KA, Lapraz F, Ballister ER, Terasaki M, Rodgers J, McDowell RJ, Johannes Girstmair J, Criswell KE, Boldogkoi M, Simpson F, Goulding D, Cormie C, Hall BK, Lucas RJ, Telford MJ | 2019 | Nucleotide sequences for *Maritigrella crozieri* r-opsin | https://www.ncbi.nlm.nih.gov/nuccore/MK598847 | GenBank, MK598847 |
| Rawlinson KA, Lapraz F, Ballister ER, Terasaki M, Rodgers J, McDowell RJ, Johannes Girstmair J, Criswell KE, Boldogkoi M, Simpson F, Goulding D, Cormie C, Hall BK, Lucas RJ, Telford MJ | 2019 | *Bugula neritina* xenopsin | https://www.ncbi.nlm.nih.gov/nuccore/BK011182 | GenBank, BK011182 |
| Rawlinson KA, Lapraz F, Ballister ER, Terasaki M, Rodgers J, McDowell RJ, Johannes Girstmair J, Criswell KE, Boldogkoi M, Simpson F, Goulding D, Cormie C, Hall BK, Lucas RJ, Telford MJ | 2019 | *Leptoplana tremellaris* xenopsin | https://www.ncbi.nlm.nih.gov/nuccore/BK011048 | GenBank, BK011048 |
| Rawlinson KA, Lapraz F, Ballister ER, Terasaki M, Rodgers J, McDowell RJ, Johannes Girstmair J, Criswell KE, Boldogkoi M, Simpson F, Goulding D, Cormie C, Hall BK, Lucas RJ, Telford MJ | 2019 | *Dendrocoelum lacteum* xenopsin | https://www.ncbi.nlm.nih.gov/nuccore/BK011049 | GenBank, BK011049 |
| Rawlinson KA, Lapraz F, Ballister ER, Terasaki M, Rodgers J, McDowell RJ, Johannes Girstmair J, Criswell KE, Boldogkoi M, Simpson F, Goulding D, Cormie C, Hall BK, Lucas RJ, Telford MJ | 2019 | *Pterosagitta draco* xenopsin 1 | https://www.ncbi.nlm.nih.gov/nuccore/BK011050 | GenBank, BK011050 |
| Rawlinson KA, Lapraz F, Ballister ER, Terasaki M, Rodgers J, McDowell RJ, Johannes Girstmair J, Criswell KE, Boldogkoi M, Simpson F, Goulding D, Cormie C, Hall BK, Lucas RJ, Telford MJ | 2019 | *Pterosagitta draco* xenopsin 2 | https://www.ncbi.nlm.nih.gov/nuccore/BK011051 | GenBank, BK011051 |
| Rawlinson KA, Lapraz F, Ballister ER, Terasaki M, Rod- | 2019 | *Prostheceraeus vittatus* rhabdomeric opsin | https://www.ncbi.nlm.nih.gov/nuccore/BK011052 | GenBank, BK011052 |

| | | | | |
|---|---|---|---|---|
| gers J, McDowell RJ, Johannes Girstmair J, Criswell KE, Boldogkoi M, Simpson F, Goulding D, Cormie C, Hall BK, Lucas RJ, Telford MJ | | | | |
| Rawlinson KA, Lapraz F, Ballister ER, Terasaki M, Rodgers J, McDowell RJ, Johannes Girstmair J, Criswell KE, Boldogkoi M, Simpson F, Goulding D, Cormie C, Hall BK, Lucas RJ, Telford MJ | 2019 | *Rhynchomesostoma rostratum* rhabdomeric opsin | https://www.ncbi.nlm.nih.gov/nuccore/BK011053 | GenBank, BK011053 |
| Rawlinson KA, Lapraz F, Ballister ER, Terasaki M, Rodgers J, McDowell RJ, Johannes Girstmair J, Criswell KE, Boldogkoi M, Simpson F, Goulding D, Cormie C, Hall BK, Lucas RJ, Telford MJ | 2019 | *Schmidtea mediterranea* rhabdomeric opsin | https://www.ncbi.nlm.nih.gov/nuccore/BK011054 | GenBank, BK011054 |
| Rawlinson KA, Lapraz F, Ballister ER, Terasaki M, Rodgers J, McDowell RJ, Johannes Girstmair J, Criswell KE, Boldogkoi M, Simpson F, Goulding D, Cormie C, Hall BK, Lucas RJ, Telford MJ | 2019 | *Stylochus ellipticus* rhabdomeric opsin | https://www.ncbi.nlm.nih.gov/nuccore/BK011055 | GenBank, BK011055 |

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
