## [Decision Letter]

Thank you for submitting your article "Extraocular, rod-like photoreceptors in a flatworm express xenopsin photopigment" for consideration by *eLife*. Your article has been reviewed by two peer reviewers, including Alejandro Sánchez Alvarado as the Reviewing Editor and Reviewer #1, and the evaluation has been overseen by K VijayRaghavan as the Senior Editor.

The reviewers have discussed the reviews with one another and the Reviewing Editor has drafted this decision to help you prepare a revised submission.

Summary:

Light sensing organs in animals are as diverse as they are intriguing in structure, morphology and molecular composition. In this carefully executed work, and clearly and elegantly written manuscript, Rawlinson et al., set out to describe the photoreceptors of the free-living flatworm polyclad *Maritigrella crozieri*. Flatworms (Platyhelminthes) are among the most diverse, biomedically important groups of invertebrates. Flatworm eyes are commonly composed of photoreceptors with rhabdomes of microvilli and pigmented cells expressing rhabdomeric opsins and conserved members of the r-opsin signaling cascade. However, the presence and nature of ciliary photoreceptors (CPRs) in these organisms remains unclear, even though xenopsins (but not c-opsins) have been shown to exist in these animals. Using a combination of transcriptomics, phylogeny, histology, and electron microscopy and cell culture cDNA expression, the authors describe and make a robust argument for the existence of CPRs in Maritigrella.

Essential revisions:

While generally positive, the reviewers have a few concerns, which must be addressed. These are as follows:

1) The reviewers consider the finding that previously undescribed ciliary structures surrounding the *Maritigrella* brain and expressing xenopsin in their phaosomes to be the key discovery being reported in this manuscript. As such, I am dissatisfied by the absence of high-resolution images of single xenopsin photoreceptors co-labeled with xenopsin expression and anti-tubulin markers to show the localization of *Maritigrella* xenopsin within the phaosome and thus support the schematic presented in Figure 3K.

2) Likewise, Xenopsins appear to be a new class of opsins and while the authors demonstrate in vitro, and for the first time, the ability of these proteins to act as light-sensing protein, it is not clear whether they are acting exclusively via a Gi/o type G protein as claimed. Because of the possibility of cross-activation of different G-proteins, the authors should either qualify these findings accordingly or demonstrate the presence of signal transduction components in the xenopsin expressing photoreceptors cells. This could be accomplished via in situ hybridizations, for example.

3) Data shown in Figure 5 seem to provide evidence of cross-reactivity and signalling promiscuity in cAMP levels (both increase and decrease) as well as calcium level changes. The authors should refrain from suggesting that the evidence they provide is sufficient to conclude exclusive coupling to Gi/o. The caveats of reconstitution experiments and the indirect evidence that these experiments provide should be clearly stated for the non-specialist.

4) The authors should provide clear evidence that the expression levels of the various opsins used in 293 cells (Figure 5) are similar. Signaling fluxes can be dramatically influenced by protein levels and comparing signals induced can be misleading if expression levels are dissimilar.

5) It is a little unclear as to why the authors describe the ciliary phaosomes as being rod-like photoreceptors. Some elaboration on the similarities between the two photoreceptor types should help clarify this. Ciliary phaosomal photoreceptors reported here are unique in many respects (as mentioned in the Discussion) and it does not appear to be necessary to be liken them to rod photoreceptors.

---

## [Author Response]

Essential revisions:While generally positive, the reviewers have a few concerns, which must be addressed. These are as follows:1) The reviewers consider the finding that previously undescribed ciliary structures surrounding the Maritigrella brain and expressing xenopsin in their phaosomes to be the key discovery being reported in this manuscript. As such, I am dissatisfied by the absence of high-resolution images of single xenopsin photoreceptors co-labeled with xenopsin expression and anti-tubulin markers to show the localization of Maritigrella xenopsin within the phaosome and thus support the schematic presented in Figure 3K.

Yes, this is a good point. Previous reports in the literature show xenopsin mRNA expression in cells with cilia and not xenopsin protein localization. Following this suggestion by the reviewers we have added a new figure (Figure 5) with two higher magnification immunofluorescence images of the xenopsin protein being expressed in the phaosomal acetylated tubulin^+^ cells of the adult worm. These images show xenopsin expression throughout the cell and on the cilia, and also show more clearly the position of the nucleus (as mentioned in minor point #9). We have also added higher magnification images of the ciliary xenopsin^+^ cells in the larval stage, and it is clear here that xenopsin and acetylated tubulin are expressed on the cilia in a cerebral eye (Figure 2C, F).

*2) Likewise, Xenopsins appear to be a new class of opsins and while the authors demonstrate* in vitro*, and for the first time, the ability of these proteins to act as light-sensing protein, it is not clear whether they are acting exclusively via a Gi/o type G protein as claimed. Because of the possibility of cross-activation of different G-proteins, the authors should either qualify these findings accordingly or demonstrate the presence of signal transduction components in the xenopsin expressing photoreceptors cells. This could be accomplished via in situ hybridizations, for example.*

The aims of our in vitro xenopsin assays in human cells were to test whether this newly classified opsin can drive changes in secondary messengers in response to light, thereby demonstrating that it can form a photopigment (i.e. bind to retinal), and couple to a Gα subunit to drive phototransduction. We have shown that *Mc* xenopsin can form a photopigment and that in this system it primarily leads to reductions in cAMP levels indicating signalling through Gα_i,_ with additional cross-coupling to at least one other pathway. We have further qualified these findings (Abstract, Results subsection “*Maritigrella* xenopsin forms a photopigment capable of sustained Gαi signalling”, and in the Discussion paragraph four) and made it more clear, that this opsin is capable of promiscuous signaling.

In carrying out this study we performed in situ hybridisation (ISH) and immunofluorescence for multiple Gα subunits to see if they are expressed in these xenopsin^+^ cells. As opsin photopigments bind to Gα subunits as the first step of the phototransduction cascade, we identified orthologs of the main Gα families (Gα_q,_ Gα_i_, Gα_o_ and Gα_s_) in the *Maritigrella* transcriptome (see phylogeny of G α subunits added as Figure 5—figure supplement 1A in the revised version) and designed RNA probes for them. Unfortunately the ISHs didn’t work. We also tried immunofluorescence using commercially available antibodies against different Gα subunits (Gα_i_-1 (R4), Gα_i_-1/2/3 (35), Gα_o_ (A2), Gα_s/olf_ (A-5), Gα_s/olf_ (C-18), Gα12 (E-12), Gα_q_/11α (C-19) Santa Cruz Biotechnology, Inc), of these only Gα_q_ showed a signal and was expressed in *r-opsin* expressing cells. During our revisions we have tried more rounds of immunofluorescence with these antibodies (this time with an antigen retrieval step) and, interestingly, we see expression of Gα_i_-1 in the xenopsin^+^ cells of the adult (Figure 5I and J) (subsection “In the adult, xenopsin is expressed in extraocular ciliary phaosomes and r-opsin is expressed in the eyes”).

The expression of xenopsin and Gαi in the ciliary phaosomes of the adult supports our observations from live cells assays which show preferential binding of xenopsin to Gαi. We also show conservation of the C-terminus of Gα subunits from *Maritigrella* and human (Figure 5—figure supplement 1B), the major site of Gα/GPCR interaction, as evidence that HEK293 cells are good test bed for testing flatworm opsin phototransduction.

In sensory neurons there are a number of cellular events occurring in parallel to environmental signalling, including cell-to-cell interactions and signalling from other nerves. Because of this, multiple Gα subunits can be expressed in a sensory neuron cell type (e.g. Gα_s_ and a Gα_i_, plus eight other Gα species are expressed in taste bud cells (Kusakabe et al., 2000). Co-localisation experiments of GPCR and Gα subunit may therefore not identify specific pathways.

Our combination of live cell assays plus immunofluorescence, however, provide two pieces of indirect evidence that suggest xenopsin may couple to Gαi in their native environment. We agree that further work is needed to determine the xenopsin signaling pathway(s) present in the native photoreceptor cells, and we have stated this in Discussion paragraph five. We have also added that to determine the phototransduction signalling cascade in the native xenopsin^+^ cell would require in vitro culture of, and secondary messenger assays directly in these phaosomal cells. Because cell culture methods for the phaosomal cells would take a long time to develop, this is, unfortunately, beyond the scope of this paper.

3) Data shown in Figure 5 seem to provide evidence of cross-reactivity and signalling promiscuity in cAMP levels (both increase and decrease) as well as calcium level changes. The authors should refrain from suggesting that the evidence they provide is sufficient to conclude exclusive coupling to Gi/o. The caveats of reconstitution experiments and the indirect evidence that these experiments provide should be clearly stated for the non-specialist.

Please see the above response for the changes we have made to clarify that we are not suggesting that xenopsin binds exclusively to Gαi.

In the Discussion (paragraph four) we discuss xenopsin cross-reactivity and signalling promiscuity. In summary, our results show that xenopsin’s primary response to light is to signal via a Gαi signal transduction cascade. Our data also indicate that *Mc* xenopsin is capable of promiscuous signaling, specifically, when Gαi is inactivated by pertussis toxin, *Mc* xenopsin acts to increase cAMP in response to light, revealing additional coupling either to Gαs or to another undefined pertussis toxin insensitive pathway. *Mc* xenopsin also drove a transient increase in intracellular Ca^2+^ but, as this response was blocked by pertussis toxin, it probably reflects crosstalk with the Gαi signaling pathway, not coupling to Gαq.

As for the caveats of this reconstitution experiment, we have strengthened the case for using HEK cells to test flatworm opsin-Ga-protein interactions by showing conservation of the C-terminal end of *Maritigrella* and human Gα subunits (Figure 5—figure supplement 1B) paragraph one subsection “*Maritigrella* xenopsin forms a photopigment capable of sustained Gαi signalling”. We also discuss that the prolonged response of Mc xenopsin in HEK cells could be because the signal termination mechanisms (e.g. GPCR kinases and β-arrestins) present in HEK293 cells may not be suitable for xenopsin. (paragraph five).

We agree that this experiment provides only indirect evidence of how xenopsin might interact with Ga proteins in its native cells and have stated this in paragraph six of subsection “*Maritigrella* xenopsin forms a photopigment capable of sustained Gαi signalling”. We also give suggestions on how to investigate xenopsin signalling in its native cell’s types (Discussion paragraph five).

4) The authors should provide clear evidence that the expression levels of the various opsins used in 293 cells (Figure 5) are similar. Signaling fluxes can be dramatically influenced by protein levels and comparing signals induced can be misleading if expression levels are dissimilar.

In light of this comment, we have carried out immunofluorescence experiments to measure expression levels of the four opsins in HEK293 cells. We have added Figure 6—figure supplement 1 showing representative images of cells expressing each opsin and control, and a plot showing measurements of fluorescent intensity for each opsin and the no opsin control. We don’t think the differences in the levels of expression alter our interpretation of the responses we see, and Mc xenopsin fluorescent intensity is significantly greater than the no opsin control (subsection “*Maritigrella* xenopsin forms a photopigment capable of sustained Gαi signalling” paragraph two).

5) It is a little unclear as to why the authors describe the ciliary phaosomes as being rod-like photoreceptors. Some elaboration on the similarities between the two photoreceptor types should help clarify this. Ciliary phaosomal photoreceptors reported here are unique in many respects (as mentioned in the Discussion) and it does not appear to be necessary to be liken them to rod photoreceptors.

In the Discussion (paragraph seven) we have expanded our comparison of flatworm ciliary phaosomes and jawed vertebrate rods. We think there are two interesting similarities between these cells; the first is the enclosure of ciliary membranes within the plasma membrane of each of these cell types, and the second is in the slow response kinetics of their respective opsins; rod opsin and xenopsin. We suggest these similarities are convergent, based on the phylogenetic distributions of enclosed ciliary membranes and the evolutionary separation of rod opsin (a ciliary opsin) and xenopsin. Gnathostome rods are unique among vertebrate ciliary photoreceptors in enclosing their ciliary membrane, our comparison with invertebrate ciliary phaosomes suggests that this trait may not be restricted to gnathostomes but may have arisen multiple times in photoreceptors in various invertebrate lineages as well. We think that the unique morphology of the ciliary phaosomes in flatworms suggest they are a flatworm novelty but that these rod-like features are worth emphasizing and discussing.